# AdaDecode: Accelerating LLM Decoding with Adaptive Layer Parallelism

**Zhepei Wei** [1]   **Wei-Lin Chen** [1]   **Xinyu Zhu** [1]   **Yu Meng** [1]

## Abstract

Large language models (LLMs) are increasingly used for long-content generation (*e.g.*, long Chain-of-Thought reasoning) where decoding efficiency becomes a critical bottleneck: Autoregressive decoding is inherently limited by its sequential token generation process, where each token must be generated before the next can be processed. This sequential dependency restricts the ability to fully leverage modern hardware's parallel processing capabilities. Existing methods like speculative decoding and layer skipping offer potential speedups but have notable drawbacks: speculative decoding relies on an auxiliary "drafter" model, which can be challenging to acquire and increases memory overhead, while layer skipping may introduce discrepancies in the generated outputs due to the missing key-value cache at skipped layers. In this work, we propose AdaDecode, which accelerates LLM decoding without requiring auxiliary models or changes to the original model parameters, while ensuring output consistency. AdaDecode leverages the insight that many tokens—particularly simple or highly-predictable ones—can accurately be generated at intermediate layers, as further layers often do not significantly alter predictions once the model reaches a certain confidence. By adaptively generating tokens at intermediate layers when confidence is high, AdaDecode enables the next token's computation to begin immediately. The remaining layer computations for early-predicted tokens are deferred and executed in parallel with subsequent tokens when needed, maximizing hardware utilization and reducing decoding latency. A final verification step ensures that early predictions match the results of standard autoregressive decoding, preserving output parity. Experiments across diverse generation tasks shows that AdaDecode consistently achieves superior decoding throughput compared to baselines with up to **1.73**×speedup, while guaranteeing output parity with standard autoregressive decoding.[1]

## 1. Introduction

The autoregressive decoding process in large language models (LLMs) is increasingly becoming a critical efficiency bottleneck for text generation (Khoshnoodi et al., 2024). As each token generation depends on previously generated ones, the inherently sequential nature of this process severely limits parallelization capabilities on modern hardware (Miao et al., 2023a). This challenge is becoming more pressing due to two recent key trends. Firstly, LLMs continue to grow exponentially in size (Kaplan et al., 2020; Hoffmann et al., 2022; DeepSeek, 2024), with billions or even trillions of parameters (Achiam et al., 2023; Zeng et al., 2023; Dubey et al., 2024; Jiang et al., 2024; Anthropic, 2024), resulting in substantially more time-consuming and resource-intensive computations at each step of token generation. Secondly, model-generated outputs are becoming progressively longer, driven by emerging applications such as long-form content creation (Pham et al., 2024; Bai et al., 2024) and long chain-of-thought (CoT) reasoning (OpenAI, 2024; Snell et al., 2024; Brown et al., 2024; Guan et al., 2025; Kimi et al., 2025; DeepSeek et al., 2025), which require massive inference steps (Huang et al., 2025). The confluence of these factors leads to significant latency in text generation, amplifying the urgent need for more efficient decoding methods.

To accelerate autoregressive decoding, two primary approaches have emerged: speculative decoding and layer skipping, as illustrated in Figure 1. Speculative decoding (Leviathan et al., 2023; Chen et al., 2023; Liu et al., 2023; Miao et al., 2023b; He et al., 2024; Huang et al., 2024; Li et al., 2024a;b; Du et al., 2024) employs a lightweight secondary model, called the "drafter", to generate candidate tokens at lower latency, which are then verified in parallel by the larger main model. However, the reliance on a

---

[1]University of Virginia. Correspondence to: Zhepei Wei <zhepei.wei@virginia.edu>, Wei-Lin Chen <wlchen@virginia.edu>, Xinyu Zhu <xinyuzhu@virginia.edu>, Yu Meng <yumeng5@virginia.edu>.

*Proceedings of the $42^{nd}$ International Conference on Machine Learning*, Vancouver, Canada. PMLR 267, 2025. Copyright 2025 by the author(s).

---

[1]Code and artifacts are available at https://github.com/weizhepei/AdaDecode.

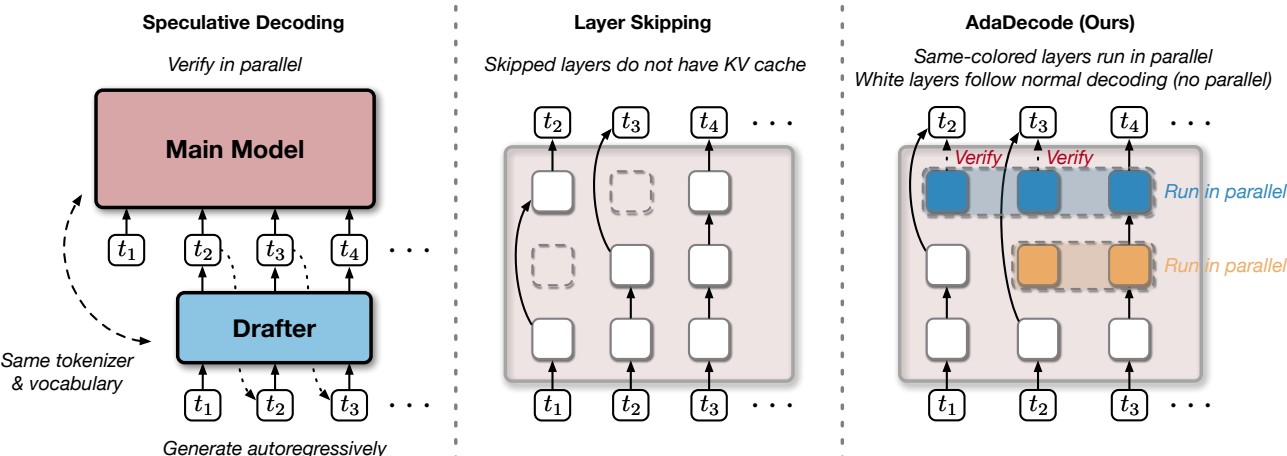

*Figure 1.* (**Left**) Speculative decoding relies on an auxiliary drafter model, leading to increased memory usage and requiring the same tokenizer and vocabulary as the main model. (**Middle**) Layer skipping bypasses certain layers, which results in missing KV cache at those layers and can introduce discrepancies in future token predictions. (**Right**) AdaDecode (Ours) accelerates decoding by *adaptively* predicting future tokens early based on confidence (*e.g.*, $t_2$ and $t_3$ are predicted from different intermediate layers), enabling earlier progression to subsequent tokens. When future token steps require KV caches from the skipped layers (due to early predictions), these missing computations are executed *in parallel* with subsequent token processing (same-colored layers). A final verification step is employed to ensure output consistency with standard autoregressive decoding.

separate drafter model increases memory overhead and can be impractical in many cases, since the drafter must share the same tokenizer and vocabulary as the main model to ensure token compatibility. Layer skipping (Huang et al., 2018; Elbayad et al., 2020; Elhoushi et al., 2024; Del Corro et al., 2023; Raposo et al., 2024; Geva et al., 2022; Din et al., 2023), in contrast, reduces computation cost by selectively bypassing certain layers during token generation. This approach often requires designing new model architectures and intricate training methods (Elhoushi et al., 2024; Raposo et al., 2024). Although effective at reducing latency, layer skipping often leads to discrepancies in output quality compared to standard autoregressive decoding (Schuster et al., 2022; Liu et al., 2024a). Specifically, skipped layers do not compute the key-value (KV) cache, which is essential for maintaining consistency in the model's predictions of future tokens. As a result, while both speculative decoding and layer skipping provide promising speedups, they come with trade-offs that pose challenges for their widespread adoption in practice.

In this work, we propose AdaDecode, a fast and accurate decoding method that accelerates autoregressive decoding through adaptive layer parallelism. AdaDecode builds on the insight that many simple and predictable tokens can be accurately generated at intermediate layers, without requiring a full pass through all model layers (Schuster et al., 2022; Del Corro et al., 2023). To optimize token generation quality at these layers, we introduce lightweight language model (LM) heads at intermediate layers, trained to minimize the KL divergence between their predictions and those of the final layer, while keeping the original model parameters frozen. Our preliminary studies (Figure 3b) show that when predictions at intermediate layers are sufficiently confident, subsequent layers are unlikely to significantly alter the output. Based on this observation, AdaDecode predicts tokens using the hidden state at intermediate layers when confidence is high, and immediately initiates processing of the next token. The remaining layers' KV cache computations are deferred and performed in parallel at future decoding steps. This parallelism mitigates the sequential bottleneck of vanilla autoregressive decoding by processing multiple tokens simultaneously, thus maximizing hardware utilization and significantly boosting overall decoding throughput. Once all KV cache computations are complete, the standard autoregressive decoding result is obtained, allowing us to verify the correctness of early predictions. Compared to speculative decoding and layer skipping, AdaDecode accelerates autoregressive decoding while maintaining output parity, without requiring auxiliary models or modifications to the original model parameters.

Our contributions are as follows: (1) We propose a lightweight intermediate-layer LM head training approach that optimizes token prediction at early layers, enabling high-confidence early predictions, with the original model parameters unchanged. Our lightweight LM head achieves performance comparable to fully parameterized LM heads from prior works (Stern et al., 2018; Cai et al., 2024), while using $31\times$ fewer parameters; (2) We introduce adaptive layer parallelism that concurrently processes multiple early-predicted tokens generated by the lightweight LM heads from different layers, significantly improving hardware utilization and decoding speed; and (3) AdaDecode demon-

*Table 1.* Comparison of AdaDecode with relevant methods. **Drafting Strategy** represents how the draft tokens are generated; **Acceleration** refers to the direction of acceleration: horizontal acceleration speedup generation across decoding time steps, while vertical acceleration focuses on speedups within each decoding time step. **Early Exiting** indicates how the model makes early predictions (*e.g.*, exiting at a fixed layer or dynamically exiting at different layers); **Verification** indicates whether a verification step is employed to correct the draft token; **Output Guarantee** represents whether the output of the method is guaranteed to be consistent with the standard autoregressive decoding—Medusa can violate the guarantee depending on its setup; **# of Params** refers to the number of parameters that need to be trained for enabling the method, which can impact its efficiency and effectiveness. The analysis of trainable parameters is based on Llama3.1-8B-Instruct, which has a total of 32 layers with a hidden size of 4096. For standard speculative decoding (SpecDec), a separate draft model typically needs to be trained, and its number of trainable parameters may vary depending on the model architecture.

| Method | Drafting Strategy | Acceleration | Early Exiting | Verification | Output Guarantee | # of Params |
|---|---|---|---|---|---|---|
| FREE (Bae et al., 2023) | Layer Skipping | Vertical | Fixed Depth | ✗ | ✗ | 8B |
| LITE (Varshney et al., 2023) | Layer Skipping | Vertical | Dynamic Depth | ✗ | ✗ | 8B |
| SpecDec (Leviathan et al., 2023) | Autoregressive | Standard | N/A | ✓ | ✓ | draft model |
| LayerSkip (Elhoushi et al., 2024) | Layer Skipping | Vertical | Fixed Depth | ✓ | ✗ | 8B |
| EESD (Liu et al., 2024b) | Layer Skipping | Vertical | Fixed Depth | ✓ | ✓ | 0.7B |
| Medusa (Cai et al., 2024) | Multi-hop Heads | Horizontal | N/A | ✓ | ✓ & ✗ | 1.5B |
| EAGLE (Li et al., 2024a) | Autoregressive | Horizontal | N/A | ✓ | ✓ | 0.3B |
| **AdaDecode (Ours)** | Layer Skipping | Vertical | Dynamic Depth | ✓ | ✓ | 48M |

strates superior decoding throughput across various text generation tasks with up to **1.73×** speedup, while ensuring identical outputs to standard autoregressive decoding.

## 2. Method: AdaDecode

In this section, we present our proposed method AdaDecode, as illustrated in Figure 2. The core concept is to start processing the initial layers of subsequent tokens early in an adaptive manner, while completing the remaining layers of all early-predicted tokens in parallel, thereby enhancing overall decoding throughput via parallel computation. We introduce the techniques for enabling early predictions using intermediate layer representations in Section 2.1, followed by a detailed explanation of our adaptive layer parallelism processing approach in Section 2.2. Note that in this work, we focus on vertical acceleration within each decoding step, which is orthogonal to methods that target horizontal acceleration across decoding time steps. Moreover, our approach preserves the exact output of standard autoregressive decoding, with consistency guaranteed by a verification step. However, not all acceleration methods that incorporate verification can ensure such output parity. A comprehensive comparison with relevant methods is provided in Table 1.

### 2.1. Lightweight LM Heads Enable Early Predictions

**Off-the-shelf LMs struggle with early predictions.** Many tokens in natural language, such as stopwords and common phrase completions, can be easily predicted and do not require the full capacity of a model for accurate generation. However, off-the-shelf LMs typically have difficulty utilizing intermediate layers for next-token prediction, as the final-layer LM head is not trained to work with intermediate-layer representations. As shown in Figure 3a, applying the

final-layer LM head to intermediate layers results in mostly low predicted probabilities for the tokens generated by the model, making early predictions with standard LMs challenging. Hence, prior research on early exiting often involves designing specific model architectures or fine-tuning existing models to enable intermediate-layer predictions (Elhoushi et al., 2024; Din et al., 2023; Del Corro et al., 2023). As a result, these acceleration methods typically fail to produce outputs consistent with those of the original off-the-shelf LMs due to changes in architecture and parameters.

**Training intermediate-layer LM heads with original model parameters frozen.** We hypothesize that many intermediate-layer representations may already contain sufficient information for predicting the next token, but the original LM head cannot directly harness their potential. To facilitate early predictions using intermediate-layer representations without further fine-tuning them, we introduce trainable LM heads $\boldsymbol{\theta}^{(i)} = \{\boldsymbol{e}_t^{(i)}\}_{t \in \mathcal{V}}$ at each candidate early prediction layer $l^{(i)}$. They take the hidden representations $\boldsymbol{h}^{(i)}$ at layer $l^{(i)}$ as *frozen* features and predict the next word distribution $p_{\boldsymbol{\theta}^{(i)}}(t|\boldsymbol{h}^{(i)})$, which are trained to approximate the last-layer prediction $p^*(t|\boldsymbol{h}^*)$ ($\boldsymbol{h}^*$ is the last-layer hidden states) by minimizing the following KL divergence loss:

$$p_{\boldsymbol{\theta}^{(i)}}(t|\boldsymbol{h}^{(i)}) = \frac{\exp\left(\boldsymbol{e}_t^{(i)} \cdot \boldsymbol{h}^{(i)}\right)}{\sum_{t' \in \mathcal{V}} \exp\left(\boldsymbol{e}_{t'}^{(i)} \cdot \boldsymbol{h}^{(i)}\right)},$$

$$\mathcal{L}(\boldsymbol{\theta}^{(i)}) = \mathrm{KL}\left(p^*(t|\boldsymbol{h}^*)\big\|p_{\boldsymbol{\theta}^{(i)}}(t|\boldsymbol{h}^{(i)})\right).$$

Since we do not update $\boldsymbol{h}^{(i)}$, the original model parameters remain unchanged, and only the newly added LM heads are trained. As demonstrated in Figure 3c, training these intermediate-layer LM heads results in good approxima-

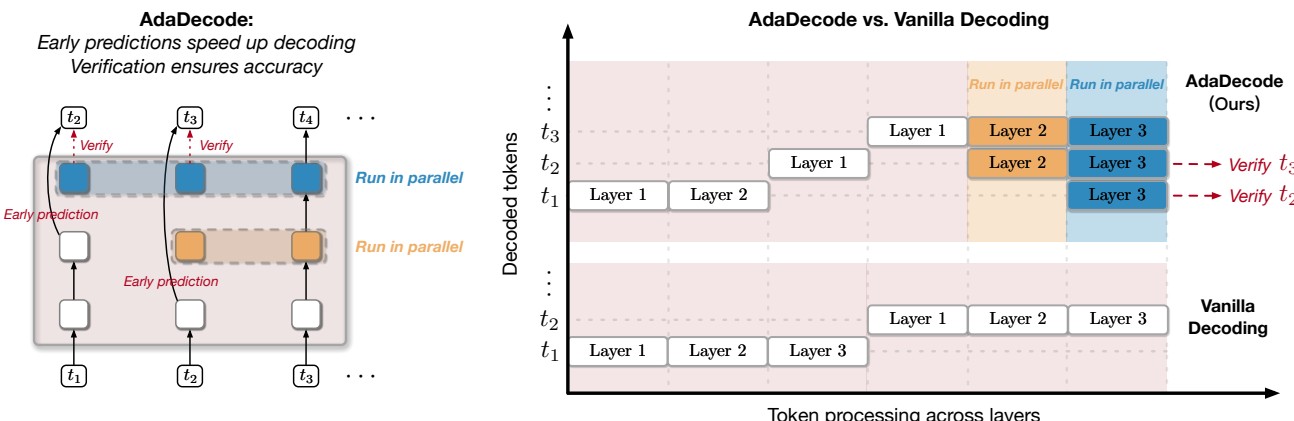

*Figure 2.* (**Left**): A simplified example demonstrating how early predictions enable parallelization. Here, $t_1$ triggers the early prediction of $t_2$ at Layer 2, and $t_2$ triggers the early prediction of $t_3$ at Layer 1. As a result, the Layer 2 computations of $t_2$ and $t_3$ can run in parallel, followed by parallelized Layer 3 computations for $t_1$, $t_2$, and $t_3$. (**Right**): Vanilla autoregressive decoding processes tokens strictly in sequence, limiting opportunities for parallelization. In contrast, AdaDecode adaptively starts processing the first few layers of the next token once the model confidently makes early predictions using intermediate LM heads. The remaining layers for all early-predicted tokens are then computed in parallel, accelerating the decoding process.

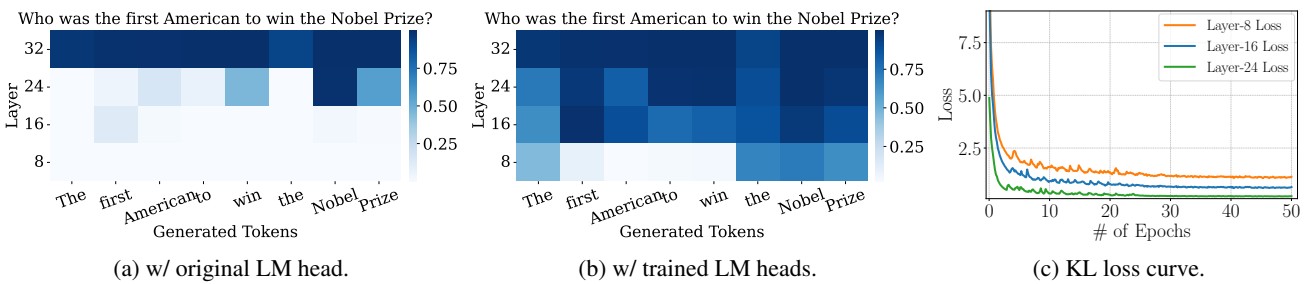

(a) w/ original LM head.  (b) w/ trained LM heads.  (c) KL loss curve.

*Figure 3.* Probabilities of model-generated tokens predicted at the 8th, 16th, 24th, and 32nd (final) layers of the fine-tuned Llama-3.1-8B-Instruct are shown using (a) the original last-layer language model (LM) head and (b) our newly introduced lightweight LM heads. These new LM heads are trained to minimize the KL divergence loss relative to the final layer predictions, while keeping all original model parameters frozen. The LM heads enable close approximation of the final predictions, as seen in (b), where many tokens have confident predictions at intermediate layers, and in (c), where the KL divergence loss relative to the final layer predictions is minimal.

tions of the last-layer outputs, as indicated by the low KL divergence loss at the end of training. This supports our hypothesis that intermediate-layer representations contain ample information for predicting the next token, and simple transformations via new LM heads can effectively extract this information. Consequently, with our trained LM heads, many tokens' predicted probabilities are notably high at intermediate layers, as shown in Figure 3b.

**Lightweight LM head implementation.** The LM heads $\boldsymbol{\theta}^{(i)} = \{\boldsymbol{e}_t^{(i)}\}_{t \in \mathcal{V}}$ are typically represented by a weight matrix $\boldsymbol{E}^{(i)} \in \mathbb{R}^{|\mathcal{V}| \times d}$ where $d$ is the model dimension size (Stern et al., 2018; Cai et al., 2024). Given the large vocabulary size of LMs, learning a separate LM head for each early prediction layer leads to a substantial increase in the number of parameters. Based on the observation that the weight matrix $\boldsymbol{E}^{(i)}$ is always applied to the hidden

states $\boldsymbol{h}^{(i)}$ to compute the probability over the vocabulary $p_{\boldsymbol{\theta}^{(i)}} = \text{Softmax}(\boldsymbol{h}^{(i)} \boldsymbol{E}^{(i)\top})$, to reduce the parameter cost, we decompose it as $\boldsymbol{E}^{(i)} = \boldsymbol{E}^* \boldsymbol{T}^{(i)}$ where $\boldsymbol{E}^*$ is the last-layer LM head weights, and $\boldsymbol{T}^{(i)} \in \mathbb{R}^{d \times d}$ is a learnable transformation matrix. As $d \ll |\mathcal{V}|$ for most LLMs, learning $\boldsymbol{T}^{(i)}$ is much more parameter-efficient than learning $\boldsymbol{E}^{(i)}$ directly. We defer the proof that learning $\boldsymbol{T}^{(i)}$ retains the full expressiveness of learning $\boldsymbol{E}^{(i)}$ to Appendix A. In practice, in an Llama3.1-8B-Instruct model (Dubey et al., 2024), each lightweight LM head $\boldsymbol{T}^{(i)}$ introduces only 16M parameters ($d = 4096$), while a full-parameterized LM head $\boldsymbol{E}^{(i)}$ would require 0.5B parameters ($|\mathcal{V}| = 128$K). To further reduce the number of trainable parameters, low-rank adaptation methods (Hu et al., 2022) can be explored, which we leave for future work.

---

**Algorithm 1:** LLM Decoding Acceleration via Adaptive Layer Parallelism (**AdaDecode**)

---

**Input:** User prompt $\boldsymbol{x} = [x_1, x_2, \ldots, x_M]$
**Parameter:** Early prediction threshold $\gamma$
**Output:** Generated output sequence $\boldsymbol{y} = [t_0, t_1, t_2, \ldots, t_N]$
KV cache $\leftarrow$ LM($\boldsymbol{x}$)      // initialize KV cache by processing user prompt
$i \leftarrow 0;\ t_i \leftarrow$ [BOS]; $\boldsymbol{y} \leftarrow [t_i]$      // initialize output sequence with [BOS] token
$\mathcal{P} \leftarrow \{\ \};\ \mathcal{P}[l] \leftarrow [\ ],\ \forall\, 1 \leq l \leq L$      // $\mathcal{P}$ tracks the list of tokens to be processed in parallel at each layer $l$
**while** $t_i \neq$ [EOS] // terminate generation upon generating EOS token **do**
    **for** $1 \leq l \leq L$ // process $t_i$ from the first to the last layer **do**
        $\mathcal{P}[l] \leftarrow \mathcal{P}[l] \oplus t_i$      // add current token to the parallel processing list at layer $l$
        update KV cache at layer $l$ by processing all tokens in $\mathcal{P}[l]$ in parallel
        $t_{i+1} \sim p_{\boldsymbol{\theta}^{(l)}}(t' | \boldsymbol{h}^{(l)})$      // sample from the intermediate LM head at layer $l$
        **if** $p_{\boldsymbol{\theta}^{(l)}}(t_{i+1} | \boldsymbol{h}^{(l)}) > \gamma$ // if the probability surpasses the threshold **then**
            $\boldsymbol{y} \leftarrow \boldsymbol{y} \oplus t_{i+1}$      // append $t_{i+1}$ to output sequence
            $\mathcal{P}[l'] \leftarrow \mathcal{P}[l'] \oplus t_i,\ \forall\, l < l' \leq L$      // add $t_i$ to the parallel processing list of all deeper layers
            **break**      // immediately start processing the next token
        **if** $l = L$ // if the final layer is reached **then**
            $\forall\, t \in \mathcal{P}[l]$, accept or reject $t$ based on Eq. (2)      // verify all tokens in $\mathcal{P}[l]$ in parallel
            $\mathcal{P}[l] \leftarrow [\ ]$      // empty $\mathcal{P}[l]$
            $\boldsymbol{y} \leftarrow \boldsymbol{y} \oplus t_{i+1}$      // remove rejected tokens, append $t_{i+1}$ to output sequence, and resume from the correct token
    $i \leftarrow i + 1$
**return** $\boldsymbol{y}$

---

## 2.2. Adaptive Layer Parallelism

**Early predictions trigger parallel processing.** As shown in Figure 3b, when a token's predicted probability is sufficiently high with intermediate LM heads (introduced in Section 2.1), subsequent layers are unlikely to change the predictions significantly. Based on this observation, we generate the next token $t$ at layer $l^{(i)}$ when its probability surpasses a predefined threshold:

$$t \sim p_{\boldsymbol{\theta}^{(i)}}(t' | \boldsymbol{h}^{(i)}) \quad \text{and} \quad p_{\boldsymbol{\theta}^{(i)}}(t | \boldsymbol{h}^{(i)}) > \gamma, \qquad (1)$$

where $\gamma$ is a hyperparameter, and any sampling strategy can be employed (*e.g.*, greedy or nucleus sampling (Holtzman et al., 2020)). This early prediction can adaptively happen at different layers (*i.e.*, any layer with an intermediate LM head) once the probability surpasses $\gamma$, allowing flexible parallel calculation of KV caches across multiple tokens. Notably, it is necessary to finish processing the subsequent layers of the early predicted token to obtain their KV cache—omitting this step would result in a missing KV cache at deeper layers, which would cause inconsistencies when computing future token representations.

**Early prediction verification.** Regardless of the threshold hyperparameter $\gamma$ set in Equation (1), there is always a possibility that the early predicted token from intermediate layers differs from the final prediction. To ensure consistency with standard autoregressive decoding, we introduce a verification step that first completes the KV cache of the remaining layers for all early-predicted tokens in parallel, and

then verifies the early predictions using a modified rejection sampling scheme (Chen et al., 2023; Leviathan et al., 2023). Specifically, we accept the early predicted token $t$, sampled from the intermediate layer's distribution $p_{\boldsymbol{\theta}^{(i)}}(t' | \boldsymbol{h}^{(i)})$, with probability:

$$\min \left\{ 1,\ \frac{p^*(t' | \boldsymbol{h}^*)}{p_{\boldsymbol{\theta}^{(i)}}(t' | \boldsymbol{h}^{(i)})} \right\}. \qquad (2)$$

If $t$ is accepted, we move on to the verification of the next early-predicted token, until all early predictions have been accepted or a token gets rejected. When an early prediction $t$ is rejected, we resample a token $t^*$ from the adjusted distribution Normalize($\max(0, p^*(t' | \boldsymbol{h}^*) - p_{\boldsymbol{\theta}^{(i)}}(t' | \boldsymbol{h}^{(i)}))$) as a replacement, remove the KV caches for tokens after $t$, and resume generation from $t^*$ onward.

Although rejecting early predictions can lead to some wasted computation, we observe in practice that this happens rarely. For example, as indicated in Figure 5c, with $\gamma = 0.85$, only about 6% of all early predictions are rejected, which supports our observation that high-confidence predictions from intermediate layers tend to align with the final predictions.

**Overall algorithm** of AdaDecode is presented in Algorithm 1.

## 3. Experimental Setups

**Backbone models and evaluation tasks.** We evaluate our method on a diverse set of text generation tasks, including

*Table 2.* Throughput (average tokens per second) and speedup comparison between AdaDecode and baseline decoding methods across three tasks, including text summarization, code generation, and mathematical reasoning. All compared methods ensure generation parity with vanilla autoregressive decoding, and the speedup results are computed relative to the benchmarks established on the vanilla setup (*i.e.*, speedup = $1\times$ for vanilla autoregressive decoding). The best performance is highlighted in **bold**.

| Method | Text Summarization (XSum) | | Code Generation (HumanEval) | | Mathematical Reasoning (GSM8K) | |
|---|---|---|---|---|---|---|
| | Throughput (Tokens/s) | Speedup (vs. Vanilla) | Throughput (Tokens/s) | Speedup (vs. Vanilla) | Throughput (Tokens/s) | Speedup (vs. Vanilla) |
| Llama3.1-8B$_{\text{INST}}$ | | | | | | |
|   Vanilla Decoding | 33.31 | $1.00\times$ | 32.58 | $1.00\times$ | 33.13 | $1.00\times$ |
|   SpecDecode (Leviathan et al., 2023) | | | | | | |
|     w/ drafter Llama3.2-1B$_{\text{INST}}$ | 35.64 | $1.07\times$ | 46.26 | $1.42\times$ | 45.38 | $1.37\times$ |
|   AdaDecode (Ours) | 38.09 | **$1.14\times$** | 49.21 | **$1.51\times$** | 49.17 | **$1.48\times$** |
| CodeLlama-13B$_{\text{INST}}$ | | | | | | |
|   Vanilla Decoding | 27.80 | $1.00\times$ | 27.55 | $1.00\times$ | 28.25 | $1.00\times$ |
|   SpecDecode (Leviathan et al., 2023) | | | | | | |
|     w/ drafter CodeLlama-7B$_{\text{INST}}$ | 26.97 | $0.97\times$ | 29.75 | $1.08\times$ | 28.53 | $1.01\times$ |
|   Self-SpecDecode (Zhang et al., 2024a) | 28.63 | $1.03\times$ | 31.40 | $1.14\times$ | 31.64 | $1.12\times$ |
|   LookAhead (Fu et al., 2024) | 33.08 | $1.19\times$ | 41.04 | $1.49\times$ | 38.42 | $1.36\times$ |
|   SWIFT (Xia et al., 2025) | 30.02 | $1.08\times$ | 36.64 | $1.33\times$ | 30.51 | $1.08\times$ |
|   AdaDecode (Ours) | 37.99 | **$1.37\times$** | 46.78 | **$1.69\times$** | 44.28 | **$1.57\times$** |
| CodeLlama-34B$_{\text{INST}}$ | | | | | | |
|   Vanilla Decoding | 17.68 | $1.00\times$ | 18.91 | $1.00\times$ | 19.16 | $1.00\times$ |
|   SpecDecode (Leviathan et al., 2023) | | | | | | |
|     w/ drafter CodeLlama-7B$_{\text{INST}}$ | 19.09 | $1.08\times$ | 26.66 | $1.41\times$ | 24.14 | $1.26\times$ |
|     w/ drafter CodeLlama-13B$_{\text{INST}}$ | 20.86 | $1.18\times$ | 23.25 | $1.23\times$ | 21.07 | $1.10\times$ |
|   Self-SpecDecode (Zhang et al., 2024a) | 18.97 | $1.07\times$ | 21.55 | $1.14\times$ | 21.84 | $1.14\times$ |
|   LookAhead (Fu et al., 2024) | 20.15 | $1.14\times$ | 26.28 | $1.39\times$ | 27.01 | $1.41\times$ |
|   SWIFT (Xia et al., 2025) | 21.92 | $1.24\times$ | 26.47 | $1.40\times$ | 25.29 | $1.32\times$ |
|   AdaDecode (Ours) | 24.35 | **$1.38\times$** | 32.78 | **$1.73\times$** | 30.68 | **$1.60\times$** |

text summarization (*i.e.*, XSum (Narayan et al., 2018)), code generation (*i.e.*, HumanEval (Chen et al., 2021)), and mathematical reasoning (*i.e.*, GSM8K (Cobbe et al., 2021)), covering a broad spectrum of language model capabilities. We employ three instruction-tuned models as the backbone, ranging from 8B to 34B parameters (*i.e.*, LLama3.1-8B-Instruct, CodeLlama-13B-Instruct, and CodeLlama-34B-Instruct), with the default chat template for decoding. Due to the page limit, please refer to Appendix B for a detailed introduction of the benchmarks.

**Baselines.** In our work, we primarily focus on comparing with efficient decoding baselines that provide output parity guarantees with standard autoregressive decoding techniques, including speculative decoding (Leviathan et al., 2023; Chen et al., 2023), LookAhead (Fu et al., 2024), self-speculative decoding (Zhang et al., 2024a) and its variant SWIFT (Xia et al., 2025). Despite their conceptual advantages, these methods have practical limitations. They often come with inherent constraints regarding model selection or necessitate task-specific model architectures. Such requirements significantly compromise their practical utility and present difficulties for real-world application—as we

will demonstrate later (§ 4.4), they can only lead to quite limited speedup or even negative speedup results compared to standard decoding unless careful hyperparameter tuning is performed. A more detailed introduction of baselines and implementation details is provided in Appendices C and D.

## 4. Results

### 4.1. Main Results

**AdaDecode consistently achieves superior inference speedup across all benchmarks.** To validate the effectiveness of our method, we compare the proposed AdaDecode with state-of-the-art efficient decoding methods across a wide range of challenging text generation tasks. As presented in Table 2, our method consistently delivers superior speedup compared to all baseline approaches regardless of the backbone model size, achieving up to $1.73\times$ speedup compared to standard autoregressive decoding.

**Speculative decoding (mostly) performs better when assisted with smaller models.** Table 2 shows that a smaller drafter model (*e.g.*, CodeLlama-7B-Instruct) generally leads

*Table 3.* Ablation study on AdaDecode. We report the performance and required additional parameters of our method by analyzing the impact of verification, adaptive layer prediction, and ablating the lightweight LM head. We report the speedup and consistency ratio (measured by string match between generation results) compared to vanilla autoregressive decoding of Llama3.1-8B-Instruct on the code generation task (HumanEval).

| Method | # New Heads | # New Params | Consistency Ratio | Speedup |
|---|---|---|---|---|
| AdaDecode | 3 | 48M | 0.996 | 1.51× |
| w/o verification | 3 | 48M | 0.652 | 1.64× |
| w/ fixed-layer early prediction | 1 | 16M | 0.996 | 1.37× |
| w/ original LM head | 0 | 0M | 0.995 | 0.84× |
| w/ mixed-domain LM head | 3 | 48M | 0.998 | 1.29× |
| w/ full-parameterized LM head | 3 | 1.5B | 0.997 | 1.49× |

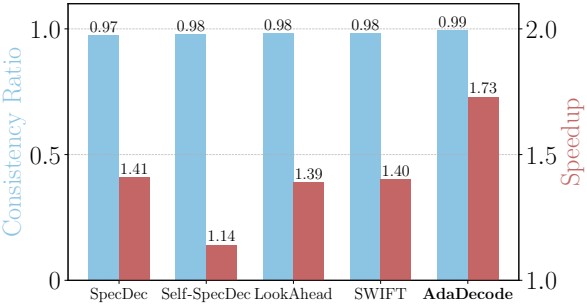

*Figure 4.* Consistency ratios and speedups of different methods vs. vanilla decoding on HumanEval with CodeLlama-34B-Instruct.

to higher speedup for SpecDecode compared to a moderate-size model (*e.g.*, CodeLlama-13B-Instruct). This is mainly because smaller drafters have fewer parameters, requiring less computation and inference time to generate draft tokens, which are then passed to a large-scale verifier (*e.g.*, CodeLlama-34B) for parallel verification and correction, significantly reducing the overall time. One notable exception is in the text summarization task, where using the smaller model as the drafter yields a lower speedup than the moderate counterpart. We speculate that this is because smaller models like CodeLlama-7B-Instruct have limited capacity to handle extremely long texts, thus producing low-quality draft tokens, which can lead to a higher rejection ratio, thereby increasing the overall latency.

**Self-SpecDecode and its variant SWIFT tend to achieve higher speedups as the model size increases**. By skipping a larger portion of intermediate layers, such methods can generate draft tokens much faster than the full model, thereby achieving significant speedup. However, this approach requires task-specific model architecture configuration to achieve meaningful speedups. This is because skipping too many layers negatively impacts the quality of the draft tokens, which will lead to a high rejection rate during verification, and consequently increase the decoding latency. On the contrary, skipping too few layers does

not yield sufficient speedup, as the performance gains stem primarily from reducing the latency of skipped layers.

**LookAhead requires surplus FLOPs to achieve meaningful speedups on larger models.** Unlike standard speculative decoding methods, LookAhead achieves higher speedup on the 13B model than on the 34B model. This is because LookAhead employs Jacobi decoding, which generates multiple disjoint n-grams in parallel within a single step. While this approach reduces the number of decoding steps, it comes at the cost of increased FLOPs. In situations where hardware FLOPs become the bottleneck, especially with large models like the 34B, the surplus FLOPs needed for parallel n-gram generation may not be satisfied, thus leading to diminished speedup.

### 4.2. Output Parity with Vanilla Decoding

**AdaDecode guarantees output parity with standard autoregressive decoding.** In principle, all compared methods in our work are designed to guarantee output consistency with standard autoregressive decoding as a result of the verification steps. However, due to numerical precision inaccuracies and potentially tied probabilities during the computation, the generation results might vary in practice and are subject to experimental environment and hardware specifications.[2] As shown in Figure 4, we empirically evaluate the output consistency ratio of all baseline methods against the vanilla decoding and report their corresponding speedup ratios on the HumanEval benchmark with CodeLlama-34B-Instruct as the backbone. Despite slightly deviating from theoretical guarantees, all of these methods empirically achieve nearly 100% consistency ratio against the standard autoregressive decoding. One potential reason is the numerical precision inaccuracies during inference, as we use FP16 precision for inference, which may cause the model to select slightly different tokens compared to standard decoding.

---

[2]https://github.com/huggingface/transformers/issues/30413

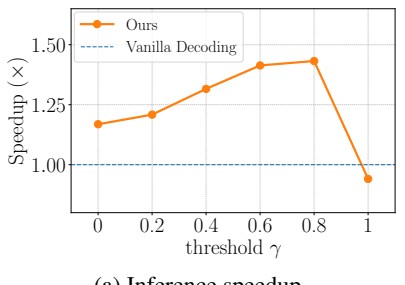
(a) Inference speedup.

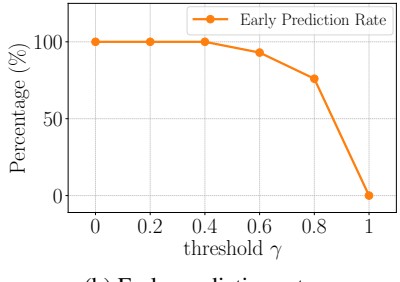
(b) Early prediction rate.

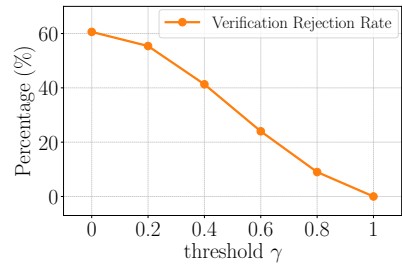
(c) Verification rejection rate.

*Figure 5.* Hyperparamter study of AdaDecode by varying the early prediction threshold $\gamma$. These figures present the evaluation results on HumanEval, including (a) the speedup curve, (b) the early prediction rate, and (c) the verification rejection rate.

## 4.3. Ablation Study

As shown in Table 3, we conduct a comprehensive ablation study to understand the effectiveness of each module in our proposed AdaDecode framework.

**The verification step is essential for ensuring output consistency.** The first row shows that removing the verification step from AdaDecode significantly reduces the consistency ratio, even though it achieves a slightly higher speedup. Without verification, there is no control over the generated tokens, which can lead to AdaDecode producing entirely different content as early predictions may not always be reliable. This results in the acceptance of potentially incorrect early predicted tokens and causes significant deviations from standard autoregressive decoding. This finding underscores the importance of the verification step in our method.

**Adaptive early prediction allows for more flexible layer parallelism.** With fixed-layer early prediction, we only need one intermediate LM head to make early predictions at a fixed layer. While this requires less memory overhead, it negatively impacts the speedup compared to the full-fledged AdaDecode that allows early prediction at different layers, demonstrating the importance of adaptive early prediction.

**Off-the-shelf original LM head for early prediction will not lead to speedup.** In this study, we replaced the trained lightweight LM head with the original final-layer LM head for early prediction. Although no additional parameters are required, it substantially compromises speedup because applying the original LM head to intermediate layers for early prediction only produces very low confidence (as shown in Figure 3a), thus slowing down the decoding process.

**Specialist LM head is better than generalist LM head.** By default, the intermediate lighter LM heads are trained for specific tasks using the training split in each benchmark. To test their effectiveness, we also train the LM heads on a mixed set of training data from all benchmarks, covering multiple domains. However, the ablation study shows that this significantly slows down the generation speed, suggesting that AdaDecode works more efficiently when the intermediate heads are trained in-domain.

**Our lightweight LM head is sufficient for early prediction compared to the full-parameterized LM head.** We also implement a variant of our method by training full-parameterized intermediate LM heads for early prediction. Despite significantly increasing the number of hyperparameters, its speedup does not gain further boost compared to our lightweight head. This echoes our hypothesis that intermediate-layer representations contain enough information for predicting the next token, and simple transformations via lightweight LM heads can effectively extract it.

## 4.4. Hyperparameter Sensitivity Study

Despite the encouraging numbers reported in the literature, replicating them is not trivial in practice, as they typically require extensive hyperparameter search. To ensure robustness, we extensively study the performance of these methods. Below, we present the results of AdaDecode, and the analysis for baselines can be found in Appendix C.

**AdaDecode is robust to hyperparameter for early predictions.** As introduced in Equation (1), the hyperparameter $\gamma$ controls early predictions and triggers adaptive layer parallelism. To study its impact, we conduct experiments with Llama3.1-8B-Instruct, and Figure 5 presents the speedup results, early prediction rate, and verification rejection rate, with varying values of $\gamma = [0, 0.2, 0.4, 0.6, 0.8, 1]$. Specifically, $\gamma = 1$ essentially means no early predictions, as no token can have a probability greater than 1, while $\gamma = 0$ triggers early predictions at every inference step. It can be observed that the early prediction rate decreases with increasing $\gamma$, while the verification rejection rate also decreases. The inference speed generally increases with the threshold unless it becomes excessively high. This finding suggests that AdaDecode works reliably as long as the threshold is within a reasonable range. We attribute this to the fine-tuned lightweight LM head, which enables early predictions with high confidence, resulting in a lower verification rejection rate (*e.g.*, around 5% when $\gamma = 0.85$) while maintaining a comparatively high early prediction rate, thus achieving meaningful speedups.

# 5. Related Work

## 5.1. Early Exiting

Early exiting enables language models to complete prediction at intermediate layers, reducing computational overhead and accelerating generation. Previous approaches achieved this by adding decision branches or language modeling heads at various depths (Teerapittayanon et al., 2016; Huang et al., 2018; Elbayad et al., 2020; Schuster et al., 2022; Yang et al., 2024; Raposo et al., 2024; Bae et al., 2023; Varshney et al., 2023). Most of these methods alter both decoding speed and output quality, while our method strictly preserves the exact output of standard autoregressive decoding. Notably, similar to our method, EESD (Liu et al., 2024b) applies early exiting with a verification step to ensure output parity. However, it is limited to fixed-depth exiting and requires an extra decoder layer and full LM head, whereas AdaDecode uses only lightweight heads and allows dynamic-depth exiting, making it more efficient and flexible. Moreover, EESD trains its auxiliary modules using cross-entropy loss and applies Thompson Sampling for stopping decisions. In contrast, AdaDecode optimizes its lightweight heads using KL divergence and employs a simple probability-thresholding strategy for draft termination. We refer the readers to Khoshnoodi et al. (2024) for a more detailed discussion.

## 5.2. Speculative Decoding

Speculative decoding (Leviathan et al., 2023; Chen et al., 2023) has emerged as an effective approach for accelerating generation. Following the standard speculative decoding approach, most works in this line attempt to draft a single chain-based sequence of draft tokens and verify them in parallel (Kim et al., 2023; Hooper et al., 2023; He et al., 2024; Fu et al., 2024; Liu et al., 2024a;b; Qin et al., 2024; Zhang et al., 2024a; Elhoushi et al., 2024; Xia et al., 2024a). Our method also falls into this category and in this work, we only compare with baselines that guarantee output parity with standard decoding. Another line of research is tree-based speculative decoding, where multiple draft sequences are generated and verified in parallel using a tree-based attention mechanism (Miao et al., 2023b; Cai et al., 2024; Ankner et al., 2024; Li et al., 2024a). While orthogonal to our main contribution, we further integrate adaptive layer parallelism to tree-based decoding and share our findings in Appendix E, which we believe merit further investigation.

# 6. Conclusion

We introduced AdaDecode, a new approach designed to accelerate LLM decoding while preserving output consistency. AdaDecode achieves this by adaptively generating the next tokens at intermediate layers using an adaptive layer par-

allelism with our newly introduced lightweight LM heads. A verification step ensures consistency with standard autoregressive outputs. Notably, it demonstrates consistent improvements in decoding throughput across various generation tasks compared to baselines, while imposing minimal constraints (*e.g.*, no auxiliary model or fine-tuning of existing model parameters required). Further discussion on limitations and future work is presented in Appendix F.

# Acknowledgments

The authors would like to thank Tianyu Gao and Mengzhou Xia from Princeton NLP group for their valuable feedback and discussions. This research was supported in part by the NVIDIA Academic Grant. We thank anonymous reviewers and area chair for their constructive and insightful comments.

# Impact Statement

Our work focuses on improving the efficiency of autoregressive LLM decoding, a well-established objective within the machine learning community. While AdaDecode does not introduce novel ethical concerns beyond those already associated with autoregressive LLMs, it remains crucial to approach the development and deployment of language models responsibly, with careful attention to issues such as fairness, bias, and potential misuse.

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

# A. Proofs

*Table 4.* Empirical validation of the last-layer LM head weights $E^*$ being full-rank. The rank (*i.e.*, number of non-zero singular values) of $E^*$ is identical to the model's hidden size, confirming that $E^*$ is indeed full-rank. Note that even the smallest singular value is significantly greater than 0, demonstrating that the matrix is full-rank not simply because of randomness or noise.

| Model | $E^*$ Shape | # of Singular Values | # of Non-Zero Singular Values | Smallest Singular Value |
|---|---|---|---|---|
| LLama-3.1-8B | $128256 \times 4096$ | 4096 | 4096 | 1.45271 |
| CodeLlama-13B | $32016 \times 5120$ | 5120 | 5120 | 1.10411 |
| CodeLlama-34B | $32000 \times 8192$ | 8192 | 8192 | 0.83576 |

**Lemma A.1.** *For any* $E^{(i)} \in \mathbb{R}^{|\mathcal{V}| \times d}$, *there exists a* $T^{(i)} \in \mathbb{R}^{d \times d}$ *such that* $E^{(i)} = E^* T^{(i)}$.

*Proof.* We first prove that $E^*$ (the last-layer LM head weights) is full-rank, and then present an explicit form of $T^{(i)}$ that satisfies $E^{(i)} = E^* T^{(i)}$. During LLM pretaining, the last-layer LM head weights $E^*$ are trained together with last layer hidden states $H^* \in \mathbb{R}^{|\mathcal{D}| \times d}$ ($|\mathcal{D}|$ is the total number of training tokens in the corpus) to learn the ground-truth next-token prediction probability $P^* \in \mathbb{R}^{|\mathcal{D}| \times |\mathcal{V}|}$:

$$P^* \approx \text{Softmax}(H^* E^{*\top}).$$

As the softmax function cannot increase matrix rank (Kanai et al., 2018), we have $\text{rank}(\text{Softmax}(H^* E^{*\top})) \leq \text{rank}(H^* E^{*\top}) \leq \min\{\text{rank}(E^*), \text{rank}(H^*)\} \leq \text{rank}(E^*)$. Thus, to achieve a good approximation of $P^*$, $\text{rank}(E^*)$ must closely match $\text{rank}(P^*)$. Given the complexity and diversity of natural language, the empirical distribution $P^*$ derived from the pretraining data is extremely high-rank (Yang et al., 2018). Therefore, to accurately model this high-rank distribution, $E^*$ must be full-rank, as indicated by Lemma 1 in Yang et al. (2018): the language modeling problem can be completely solved (*i.e.*, achieve a 0 loss) if $\text{rank}(P) < d$. However, this is practically infeasible due to the inherent complexity of natural language, suggesting that $\text{rank}(P) > d$. In addition, we also provide an empirical validation in Table 4, which further confirms that $E^*$ is indeed full-rank.

Given $E^*$ being full-rank, $U := E^{*\top} E^*$ is invertible. We can define $T^{(i)}$ as $T^{(i)} = U^{-1} E^{*\top} E^{(i)}$, then:

$$E^* T^{(i)} = E^* U^{-1} E^{*\top} E^{(i)} = \left( \underbrace{E^* (E^{*\top} E^*)^{-1} E^{*\top}}_{=P} \right) E^{(i)}$$

Note that $P = E^* (E^{*\top} E^*)^{-1} E^{*\top}$ is the projection matrix onto the column space of $E^*$ (as $P^2 = P$). Since $E^*$ is full rank, $E^{(i)}$ lies in the column space of $E^*$. Therefore, applying $P$ to $E^{(i)}$ gives $E^{(i)}$ itself, confirming that $E^{(i)}$ can be expressed as $E^* T^{(i)}$.

$\square$

# B. Benchmark Details

We evaluate our method on a diverse set of text generation tasks, including text summarization, code generation, and mathematical reasoning, covering a broad spectrum of language model capabilities.

**Text summarization.** For text summarization, we use the widely adopted extreme summarization (XSum) dataset (Narayan et al., 2018), where the models are prompted to produce a single-sentence summary of a news article, testing their ability to identify and precisely summarize the most salient information in a coherent sentence. Following previous works (Zhang et al., 2024a), we randomly sample 1K instances from the test split for evaluation, and 10K instances from the training split for training the lightweight LM head.

**Code generation.** For code generation, we evaluate our method on the HumanEval (Chen et al., 2021) benchmark, which assesses Python programming skills through a variety of coding problems, ranging from basic tasks to complex problem-solving challenges. Since the standard HumanEval benchmark does not provide a training set, we use the entire MBPP (Austin et al., 2021) dataset for training, which contains a set of crowd-sourced Python programming problems designed to be solvable by entry-level programmers, covering programming fundamentals and standard library functionality. This results in a total of 974 training samples and 164 test samples for this task.

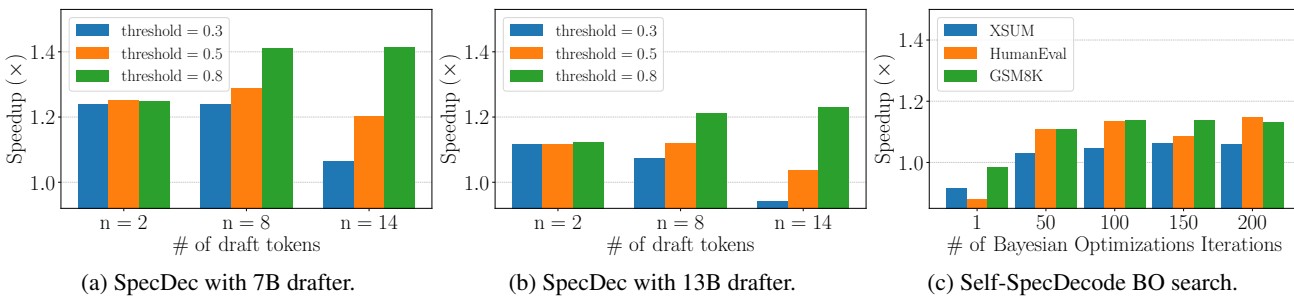

*Figure 6.* Hyperparameter study of baseline models with CodeLlama-34B as the backbone model. (a) SpecDecode with CodeLlama-7B as drafter. (b) SpecDecode with CodeLlama-13B as drafter. (c) Self-SpecDecode optimized with Bayesian optimization (BO) search.

**Mathmatical reasoning.** We use GSM8K (Cobbe et al., 2021) as the benchmark for mathematical reasoning, which contains diverse grade-school math word problems created by human problem writers. The dataset consists of 7.5K training problems and 1K test problems. These problems typically require multiple reasoning steps to solve and involve performing a sequence of basic arithmetic operations (such as addition and subtraction) to arrive at the final answer. The goal of this task is specifically to evaluate the LLM's ability in multi-step mathematical reasoning.

## C. Baselines

By default, we employ three instruction-tuned models as the backbone, ranging from 8B to 34B parameters (*i.e.*, LLama3.1-8B-Instruct, CodeLlama-13B-Instruct, and CodeLlama-34B-Instruct), with the default chat template for all compared methods. Since LookAhead is a training-free method, we directly use the optimized hyperparameter configuration in their released codebase to reproduce the results of backbone models on three benchmarks. For SpecDecode, we consider three configurations: (1) Llama3.1-8B-Instruct as the main model (*i.e.*, verifier) with Llama3.2-1B-Instruct as the assistant model (*i.e.*, drafter), (2) CodeLlama-13B-Instruct as the verifier with CodeLlama-7B-Instruct as the drafter, and (3) CodeLlama-34B-Instruct as the verifier with CodeLlama-13B-Instruct/CodeLlama-7B-Instruct as the drafter.

For Self-SpecDecode and its extension SWIFT, we adopt two backbone models: (1) CodeLlama-13B-Instruct, and (2) CodeLlama-34B-Instruct since their released codebase does not support recent models like the Llama3 or newer series. Following Zhang et al. (2024a), we adopt an adaptive confidence threshold strategy and set the initial threshold $\gamma^0 = 0.6$, the max number of draft token $K = 12$, and the max number of generated token $T = 512$. We run the Bayesian optimization search for 200 iterations to determine skipped layers for configuring the drafter model. We use 4 instances randomly sampled from the training set of each task for the Bayesian optimization search as suggested in the original implementation. More implementation details regarding training and inference can be found in Appendix D.

### C.1. Hyperparameter Sensitivity Study on Baseline Methods

**SpecDecode is sensitive to hyper-parameter configurations.** Figure 6 presents the hyperparameter study for SpecDecode, indicating that it is challenging to achieve consistent speedup without careful tuning. In this study, we use CodeLlama-34B-Instruct as the backbone model, and performed a comprehensive hyperparameter search on the HumanEval benchmark, with CodeLlama-7B-Instruct and CodeLlama-13B-Instruct as the assistant models, respectively. We explored various configurations with different max number of draft tokens $n = [2, 8, 14]$ and confidence threshold = $[0.3, 0.5, 0.8]$. Figure 6a and Figure 6b reveal that significant speedup variance exists across different settings. Moreover, the variance becomes even larger when the confidence threshold shifts. Notably, when $n = 14$ and the threshold is 0.3, the speedup of 13B drafter is merely 0.944, which is even worse than standard decoding. These findings highlight that careful tuning hyperparameters is essential to optimize SpecDec's speedup performance and avoid potential slowdowns.

**Self-SpecDecode and its variant SWIFT requires task-specific architecture for optimized speedup performance**. Recent works like Self-SpecDecode have been proposed as a training-free acceleration method to be employed in a plug-and-play manner. However, it requires an additional Bayesian optimization (BO) process to first obtain a set of layers to skip (as the drafter) before it can be adopted for inference. The time-consuming BO process greatly limits its practicality, moreover, it requires a set of examples as validation data to select the desired drafter. To further investigate the effect of this search process, we optimize CodeLlama-34B-Instruct with different numbers of BO iterations, as shown in Figure 6c. In general, more number of iterations could lead to improved evaluation performance. Yet, on HumanEval, it demonstrates a

decrease in performance in the process. This implies scaling the number of iterations does not monotonically reflect better results, and the iteration process might require careful tuning to select an optimized drafter for inference.

## D. Implementation Details

**Training details.** The lightweight LM heads in our method are trained through full-parameter fine-tuning using the alignment-handbook repository[3] with 8×Nvidia A100 GPUs. Specifically, we utilize DeepSpeed ZeRO-3 (Rajbhandari et al., 2020) along with FlashAttention (Dao, 2024) for distributed training, and we enable BF16 mixed precision training to enhance training efficiency. We generate on-policy data to train the lightweight LM heads by prompting the off-the-shelf Llama3.1-8B-Instruct or CodeLlama-34B-Instruct to produce responses using greedy sampling for each benchmark. By default, our models are trained using the Adam optimizer (Kingma & Ba, 2014) for 100 epochs, with a batch size of 128, a learning rate of 5e-3, and a cosine learning rate schedule with 3% warmup steps.

**Inference details.** During inference, we adopt the zero-shot evaluation by directly prompting the model to generate responses and apply the corresponding chat templates to format the prompts, as all backbone models used in our work are instruction-tuned versions. The framework is implemented using the HuggingFace Transformers library,[4] and we set the sampling temperature to zero in all methods for a reproducible comparison. Following Zhang et al. (2024a), the maximum number of new tokens is set to 512. The threshold $\gamma$ in Equation (1) is set to 0.75, and to ensure a balance between the early prediction rate and the rejection rate, we limit the maximum number of early predictions to 5.

## E. Integration with Tree-based Speculative Decoding

**Naive combination of adaptive layer parallelism and tree-based decoding does not yield performance improvement.** As discussed in Section 5, tree-based speculative decoding methods (Miao et al., 2023b; Cai et al., 2024) and early exiting techniques are orthogonal to each other: the former accelerates decoding horizontally by generating multiple tokens across time steps, while the latter reduces per-token computation through vertical acceleration via adaptive layer parallelism enabled by early predictions. These approaches address different bottlenecks in the decoding process.

To explore the effect of adaptive layer parallelism in the context of tree-based decoding, we augment Llama3.1-8B-Instruct by inserting and training two additional lightweight LM heads at the final layer (layer 32) and three lightweight heads at each of two intermediate layers (layers 16 and 24). These extra LM heads support multi-hop next-token predictions (*i.e.*, one-hop, two-hop, and three-hop), and we construct draft trees following a predefined structure used in Medusa (Cai et al., 2024). To establish the vanilla tree-based decoding baseline, only the heads at the final layer are used to generate multi-hop predictions, while intermediate-layer heads are inactive. Like Medusa, the original LM head (for one-hop prediction) at the final layer is used to perform verification and determine the final output tokens.

In contrast, to evaluate the impact of adaptive layer parallelism, we further allow the model to generate draft trees at intermediate layers using the lightweight heads. Draft tokens produced at shallower layers can be verified by subsequent deeper layers and ultimately finalized by the last layer. Although this introduces additional verification costs compared to the vanilla tree-based decoding baseline, our intuition is that if early draft trees are reasonably reliable with high acceptance rates, the extra verification cost could be offset by reduced computation from early exits.

We train and evaluate this model on the HumanEval task, and the results of this exploratory study demonstrate that combining vertical acceleration (via adaptive layer parallelism) with horizontal acceleration (via tree-based speculative decoding) generally leads to greater speedups than using vertical acceleration alone. However, it is unexpected that allowing that model to generate draft trees at intermediate layers does not appear to outperform the vanilla tree-based method that generates all draft tokens only at the final layer. **In other words, the naive combination of vertical and horizontal acceleration does not yield improvements over horizontal acceleration alone**, suggesting that a more thoughtful integration strategy is needed to fully exploit their complementary strengths. We hypothesize several possible reasons for this observation. First, draft tokens generated from intermediate layers are often less reliable, resulting in a higher rejection rate during final-layer verification and ultimately diminishing the potential efficiency gains. Second, different layers may favor different tree structures—shallower layers, due to their lower confidence, may benefit from narrower trees with fewer nodes at deeper hops (*i.e.*, making fewer two-hop or three-hop predictions), while deeper layers can support wider trees with more aggressive

---

[3]https://github.com/huggingface/alignment-handbook
[4]https://github.com/huggingface/transformers

exploration. Therefore, naively applying a predefined tree structure across all layers may lead to suboptimal results. These findings suggest that, while combining vertical and horizontal acceleration holds promise, further research is needed to address these challenges and fully realize the potential of their integration.

## F. Limitations and Future Work

**Limitations.**   Our work focuses on accelerating LM decoding without directly improving the quality of the outputs, so AdaDecode may encounter similar issues as general LM decoding, such as hallucinations (Huang et al., 2023; Liu et al., 2025) and generating unsafe content (Zhang et al., 2024b; Yu et al., 2024; Chen et al., 2024). Additionally, while AdaDecode consistently accelerates standard autoregressive decoding, it may occasionally incur higher FLOPs, particularly when an early-predicted token does not match the gold token, necessitating a reversion. However, such cases are rare in practice, and we find that strict parity with standard autoregressive decoding is not always necessary. The early-predicted tokens, even when they differ from the final prediction, are typically still meaningful. Therefore, one might consider relaxing strict consistency requirements to avoid unnecessary computational waste.

**Future work.**   In the future, we plan to integrate AdaDecode with other efficiency-enhancing techniques like model pruning and quantization. Model pruning (Xia et al., 2024b) offers the potential to reduce model size and parameter space by identifying and removing less critical weights or neurons. When combined with AdaDecode, pruning could lead to even faster decoding times and lower memory usage without significantly impacting performance. Similarly, AdaDecode can also be seamlessly integrated with quantization (Dettmers et al., 2023), which reduces the precision of model weights and activations, substantially lowering memory and compute requirements. It would also be interesting to apply the layer-dropping training strategy, as employed in LayerSkip (Elhoushi et al., 2024), to our method and investigate the trade-offs between training efficiency and model performance. Additionally, exploring the Thompson sampling strategy used in EESD (Liu et al., 2024b) as an alternative to our current threshold-based stopping criterion may also provide complementary benefits and enhance overall flexibility. On the other hand, as discussed in Appendix E, we aim to explore effective strategies for integrating tree-based methods (Cai et al., 2024; Miao et al., 2023b) with AdaDecode by generating multiple candidate tokens at intermediate layers of the model, which may yield even better speedups. These directions could be particularly valuable for efficient inference on mobile and edge devices.

