# OpenReview forum: "AdaDecode: Accelerating LLM Decoding with Adaptive Layer Parallelism"
_ICML.cc/2025/Conference — ICML 2025 poster_

### Official Review · Reviewer_5F7j · 2025-03-12

**Overall Recommendation:** 2

**Summary:**

This paper introduces AdaDecode, a self-speculative decoding method with an early exiting mechanism. Based on empirical findings that many simple and predictable tokens can be accurately generated at intermediate transformer layers, the authors propose three key contributions. First, they introduce a lightweight intermediate-layer LM head training approach that enables high-confidence early predictions without modifying the original model parameters. This lightweight head achieves performance comparable to fully parameterized LM heads from prior works, and is tuned to minimize KL divergence with final layer outputs. Second, they develop adaptive layer parallelism that concurrently processes multiple early-predicted tokens generated by the lightweight LM heads from different layers, significantly improving hardware utilization and decoding speed. Third, by incorporating early exiting with self-speculative decoding, AdaDecode enables KV-cache sharing between draft and verification stages, reducing computational resources while maintaining output consistency.

**Claims And Evidence:**

- The paper's core claims about the effectiveness of lightweight intermediate-layer LM heads and adaptive layer parallelism are generally well-supported by the presented evidence:
    - The proposed method of using KL divergence between final layer and intermediate layer output distributions is well-motivated and effectively implemented. This approach successfully calibrates log probability between draft and verification stages, as implicitly demonstrated in the experiments about relation between **γ** and rejection rate (Figure 5(c))
    - The claim of achieving comparable performance with lightweight head is supported by comprehensive experimental results across multiple model architectures (e.g. Llama3.1-8B, CodeLlama-13B/34B-inst) and datasets (e.g. XSum, HumanEval, GSM8K)
- However, there are concerns about the robustness and justification for the early exiting layer selection policy:
    - The authors select three intermediate layers for early exiting (8th, 16th, 24th) across different model architectures without providing sufficient analysis justifying this specific configuration. While their experiments with LLaMA3.1-8B, CodeLLama-13B, and 34B show the approach works well, the paper lacks ablation studies or theoretical analysis explaining why selecting three layers works well.
    - Previous work in layer skipping or early exiting typically incorporates additional fine-tuning or specialized architectures to determine optimal exiting points. The absence of a more sophisticated approach to layer selection raises questions about whether the reported speedups represent the maximum potential of the method.
    - A more thorough analysis of different layer configurations would strengthen the paper's claims and potentially provide insights for adapting AdaDecode to other model architectures or domains in future work.

**Essential References Not Discussed:**

The paper is missing a critical citation related to its core contributions. EESD (Early-exiting Speculative Decoding), published in ACL 2024 Findings presents remarkably similar techniques and should be acknowledged.

[1] Liu, Jiahao, et al. "Speculative decoding via early-exiting for faster llm inference with thompson sampling control mechanism." arXiv preprint arXiv:2406.03853 (2024).

**Experimental Designs Or Analyses:**

I have reviewed the experimental designs and analyses in the paper, with particular focus on their methodological soundness:

The ablation studies examining the robustness of AdaDecode to the hyperparameter **γ** (used as a threshold for the drafting stage) are well-designed and provide valuable insights into the method's stability. The authors appropriately vary this key parameter and demonstrate consistent performance across a reasonable range of values, which strengthens confidence in the practical applicability of the method.

The overall experimental framework is methodologically sound, with the authors:
- Testing on multiple model architectures (LLaMA3.1-8B, CodeLLama-13B, 34B) to demonstrate generalizability
- Evaluating on diverse tasks including summarization, mathematical reasoning, and code generation
- Providing appropriate baselines for comparison including vanilla decoding and alternative speculative methods
- Measuring both throughput improvement and output consistency to balance speed and quality

However we do have some suggestion that could provide more insight for the future readers of this paper.

1. Context Length Impact on Rejection Rates

While the paper demonstrates effectiveness on standard benchmarks, it lacks analysis of how rejection rates scale with context length. This is critical because:

- Draft quality and acceptance rates often degrade with longer contexts due to increased model strain
- Smaller models like CodeLlama-7B may struggle to maintain draft quality for long-context inputs (as mentioned in paper)

Suggested Additions:

- Experiments measuring acceptance rates across varied context lengths (e.g., 4K → 32K → 128K tokens)
- Comparison of rejection ratios between AdaDecode and baselines under long-context scenarios
- Analysis of whether lightweight LM heads mitigate long-context challenges better than conventional drafting approaches

2. Maximum New Token Length Analysis

The current experiments only test up to 512 new tokens, which may not reveal efficiency patterns for different generation demands:

Suggested Additions:

Comparative tests with max_token ∈ {256, 512, 1024} to evaluate:
- Throughput consistency across generation lengths
- Rejection rate trajectories over extended sequences
- Memory utilization patterns during prolonged generation

These additions would better demonstrate AdaDecode's robustness across practical deployment scenarios while addressing inherent challenges in speculative decoding systems.

**Methods And Evaluation Criteria:**

- The paper's proposed methods are technically sound and well-aligned with the goal of improving inference efficiency through speculative decoding:
    - The lightweight intermediate-layer LM head approach is a practical solution that avoids modifying original model parameters while enabling efficient early predictions.
    - The adaptive layer parallelism technique represents a meaningful advancement in hardware utilization for speculative decoding systems.
- Regarding evaluation, the authors employ standard metrics for the field:
    - The paper uses conventional evaluation metrics for speculative decoding: throughput (Tokens/s) and relative speedup compared to vanilla decoding. These metrics do allow for basic comparisons with prior work in the field.
    - The evaluation across different model architectures (LLaMA3.1-8B, CodeLLama-13B, 34B) and diverse tasks provides good coverage of practical applications.
- However, there is a notable limitation in the reproducibility of evaluation results:
    - While throughput and speedup are standard metrics, they are hardware-dependent, which can make cross-study comparisons challenging.
    - The paper would benefit from more comprehensive evaluation details such as: detailed hardware specifications for reproducibility, acceptance rates for speculative tokens across different scenarios, memory utilization statistics, and perhaps theoretical computation reduction metrics. These additional measures would provide a more complete picture of the method's efficiency beyond raw throughput numbers, especially for researchers working with different hardware configurations.
- Also, the comparison methodology with previous state-of-the-art speculative decoding methods presents a significant limitation in the paper's evaluation:
    - According to Supplement section C, the authors appear to have compared AdaDecode with previous methods like Self-SpecDecode using configurations that differ from those reported in the original papers. This approach undermines fair comparison, which is a cornerstone of reproducible machine learning research
    - The specific example of comparing with Self-SpecDecode on XSum using LLaMa-3.1-7B is particularly problematic since this configuration wasn't reported in the original Self-SpecDecode paper. This is especially concerning because Self-SpecDecode utilizes Bayesian optimization for hyperparameter tuning, which is highly sensitive to initial conditions and configuration settings.
    - Hyperparameter tuning significantly impacts model performance and reproducibility. When comparing methods that rely on different tuning approaches, ensuring consistent evaluation conditions becomes critical.
- For improved reproducibility, the authors should:
    - Provide the exact scripts used for Bayesian optimization in their comparisons, including random seeds to ensure deterministic results.
    - Alternatively, conduct experiments using the exact model architectures and datasets reported in the original papers they compare against.

**Other Comments Or Suggestions:**

Algorithm 1 provides a helpful overview of how AdaDecode works, which aids reviewer understanding. However, the verification stage is not adequately presented in the algorithm.

Specifically, the algorithm should explicitly express how KV cache management is handled between draft and verification stages. There should be clear notation or steps indicating that the KV cache values generated for accepted draft tokens are preserved and reused during the verification stage, while KV cache values for rejected drafts are discarded. This KV cache reuse is a critical aspect of the method's efficiency gains, as it prevents redundant computation for tokens that pass verification.

This clarification would improve the algorithm's completeness and better highlight one of the key advantages of the AdaDecode approach - the ability to share computation between draft and verification stages through KV cache reuse. Without this explicit indication, readers might miss this important implementation detail that contributes significantly to the method's performance benefits.

**Other Strengths And Weaknesses:**

- Although the paper's contributions are well aligned with its motivation, there is concern that these contributions may overlap significantly with concurrent works. For instance, EESD[1] proposes a self-speculative decoding framework with early exiting layers that bears notable similarities to AdaDecode:
    - Both methods train early exiting layers while keeping the original model's parameters fixed
    - EESD[1] uses self-distillation which performs similarly to the KL divergence approach in AdaDecode
    - Both methods highlight that KV cache created at the draft stage is reusable during the verification stage
- Given these substantial overlaps in core technical approaches, the authors should more clearly articulate what contributions of AdaDecode are orthogonal to or extend beyond those of EESD[1]. This clarification would help position the paper's contribution.


[1] Liu, Jiahao, et al. "Speculative decoding via early-exiting for faster llm inference with thompson sampling control mechanism." arXiv preprint arXiv:2406.03853 (2024).

**Questions For Authors:**

1. The paper presents experiments with pre-selected early exiting layers across different model architectures. Could you provide preliminary experimental results that justified this specific selection? Alternatively, could you present additional experiments demonstrating that AdaDecode's performance is robust to different choices of early exiting layers? This would strengthen the generalizability claim of your approach.

2. In the comparison with baseline methods like Self-SpecDecode, did you conduct experiments using the exact configurations reported in the original papers? If modifications were necessary, could you provide more details about how you ensured fair comparison? This would address concerns about reproducibility and the validity of reported performance improvements.

3. While the benchmark datasets used are reasonably diverse, they may not fully represent challenging scenarios where early exiting might underperform. Specifically, tasks requiring processing of long contexts or generating open-ended responses (e.g. ELI5, NarrativeQA) might present different dynamics for speculative decoding. Have you evaluated AdaDecode on such tasks, and if so, could you share those results?

4. Could you elaborate on how AdaDecode is orthogonal to recently published Early-exiting Speculative Decoding (EESD) in ACL 2024 Findings? EESD similarly uses early-exiting structures and self-distillation (comparable to your KL divergence approach), while employing a Thompson Sampling Control Mechanism. What specific technical innovations in AdaDecode differentiate it from EESD's approach, and how might these differences contribute to performance improvements?

ELI5 : https://facebookresearch.github.io/ELI5/
Narrative QA : https://huggingface.co/datasets/deepmind/narrativeqa_manual

**Relation To Broader Scientific Literature:**

AdaDecode's lightweight LM head approach builds upon previous early exit methods, but distinguishes itself by offering a more parameter-efficient implementation. While prior works like LayerSkip[1] also explore early exiting with shared LM heads, AdaDecode's specific parameter efficient LM head appears to be orthogonal contribution. As author mentioned future works for using LoRA instead of lightweight LM head is also expected approach.

The paper's approach of combining multiple early exiting layers with self-speculative decoding connects to concurrent works. AdaDecode's adaptive layer parallelism for concurrent processing of multiple early-predicted tokens represents a potentially orthogonal optimization approach.

The reuse of KV-cache between drafting and verification stages appears in several recent works including LayerSkip[1], EESD[2]

[1] Elhoushi, Mostafa, et al. "LayerSkip: Enabling early exit inference and self-speculative decoding." arXiv preprint arXiv:2404.16710 (2024).
[2] Liu, Jiahao, et al. "Speculative decoding via early-exiting for faster llm inference with thompson sampling control mechanism." arXiv preprint arXiv:2406.03853 (2024).

**Theoretical Claims:**

The paper provides theoretical justification for its lightweight LM head approach in Supplement A. Upon examination, the proof focuses on establishing the mathematical existence of a transformation matrix T that enables parameter-efficient implementation of intermediate layer heads. The mathematical derivations appear correct and thoroughly support the claims made in the main paper.

---

> ### Author Rebuttal · Authors · 2025-04-01
>
> We sincerely appreciate the reviewer's insightful feedback.
>
> **[Q1]: Could you present additional experiments demonstrating that AdaDecode's performance is robust to different choices of early exiting layers?**
>
> **[A1]**: Please refer to Table 1 in this [PDF](https://anonymous.4open.science/r/AdaDecode-ICML2025-132E/rebuttal/To_Reviewer_5F7j.pdf). The results show that AdaDecode achieves consistent speedups across different layer configurations.
>
> **[Q2]: The paper is missing a critical citation Early-exiting Speculative Decoding (EESD). Could you elaborate on what specific technical innovations in AdaDecode differentiate it from EESD's approach?**
>
> **[A2]**:  We have cited EESD in our original submission (Line 565). Below is a detailed comparison.
>
> - **Efficiency**: To enable reliable early exiting, EESD requires an extra decoder layer and a full LM head (vocab size × hidden size), adding ~0.7B trainable parameters.  In contrast, AdaDecode introduces only three lightweight LM heads (hidden size × hidden size), totaling just 48M additional parameters, making it significantly more efficient.
>
> - **Flexibility**: While both methods use early exiting for drafting, AdaDecode supports dynamic-depth early exiting that allows adaptive layer parallelism, whereas EESD is restricted to fixed-depth early exiting.
>
> - **Training and stopping strategy**: EESD trains its additional modules using cross-entropy loss and employs Thompson Sampling for stopping decisions. AdaDecode instead optimizes lightweight heads via KL divergence and uses probability-thresholding for termination.
>
> **[Q3]: The paper would benefit from more comprehensive evaluation details such as: (1) detailed hardware specifications for reproducibility, (2) acceptance rates for speculative tokens across different scenarios, (3) memory utilization statistics, and perhaps (4) theoretical computation reduction metrics.**
>
>
> **[A3]**:  Please refer to Appendix D and Figure 5 for a detailed discussion on (1) and (2). For (3), we maximize the memory utilization during training (~80G per GPU), and the memory utilization during inference varies depending on context length and batch size, starting from ~18G with FP16 precision.
>
> For (4), theoretical computation reduction depends on the number of total generated tokens and number of accurate early predictions. Let $\alpha$ be the fraction of layers needed for early exiting. Given $N$ tokens and per-token latency $T$, vanilla decoding takes $NT$,  while AdaDecode requires $T (1+ \alpha (N-1))$. As $N \to \infty$, the theoretical speedup is $1/\alpha$. In practice, the value of  $\alpha$ can vary based on model size. For instance, in a 48-layer model (e.g., CodeLlama-34B), exiting at layer 12 ideally gives $\alpha \sim 0.25$, leading to a potential $4\times$ speedup.
>
> **[Q4]: In the comparison with baseline methods like Self-SpecDecode, did you conduct experiments using the exact configurations reported in the original papers? If modifications were necessary, could you provide more details about how you ensured fair comparison?**
>
>
> **[A4]**: Yes, we use **exactly the same configuration** as Self-SpecDecode for CodeLlama-13B, as it was also adopted in the original paper. However, since they did not report results for LLaMA-3.1-8B, we used their Bayesian optimization script to search extensively for the best hyperparameters (Appendix C), ensuring a fair comparison.
>
> **[Q5]: Provide the exact scripts used for Bayesian optimization in their comparisons, including random seeds to ensure deterministic results.**
>
> **[A5]**: We used the same Bayesian optimization script from the official repo of Self-SpecDecode: https://github.com/dilab-zju/self-speculative-decoding/blob/main/search.ipynb.
>
> **[Q6]: Context Length Impact on Rejection Rates.**
>
>
> **[A6]**:  Our AdaDecode exhibits consistent speedups across different scenarios with varying context length. Please refer to Table 2 in this [PDF](https://anonymous.4open.science/r/AdaDecode-ICML2025-132E/rebuttal/To_Reviewer_5F7j.pdf).
>
> **[Q7]: Maximum New Token Length Analysis.**
>
> **[A7]**:  Please refer to Table 3 in the same PDF, which confirms our method’s effectiveness with different max new token lengths.
>
> **[Q8]: Algorithm 1 should explicitly express how KV cache management is handled between draft and verification stages.**
>
> **[A8]**: Indeed, KV cache management is handled internally in Line 238. If any rejection occurs, the KV cache of discarded tokens will be cleared, and the entire KV cache length is truncated to the last verified token. This ensures that generation resumes from the correct token, as described in Section 2.2. We will update the annotations in Algorithm 1 to make this clearer.
>
> If you find our response satisfactory, we would be grateful if you could consider raising your score. Thanks again for your time!

---

### Official Review · Reviewer_KXLJ · 2025-03-13

**Overall Recommendation:** 5

**Summary:**

The authors present AdaDecode, a methodology to accelerate decoding without auxiliary models or modification of the model. The proposed approach adaptively predicts tokens from an intermediate layer based on confidence, using a set of additional lightweight LM heads whose predictions are verified using a rejection sampling scheme.
AdaDecode retains output consistency and achieves a speed-up of 1.73x over vanilla decoding, outperforming four baselines: speculative decoding, self-speculative decoding, LookAhead, and SWIFT.
Moreover, the authors also include ablation studies and hyperparameter sensitivity experiments to validate design decisions and better understand the impact of the algorithm's different components.

**Claims And Evidence:**

The author claims are factual and revolve around algorithm speed-up and output consistency. Both claims are clearly backed up by the presented experiments.

**Essential References Not Discussed:**

I'm not aware of any essential related work not referenced.

**Experimental Designs Or Analyses:**

The paper and appendix clearly describe experimental designs and analyses.
I tried to run the code, but at the current stage, some needed artifacts are missing (e.g., adadecode/llama_3.1_8b_instruct).

**Methods And Evaluation Criteria:**

The methodology used to evaluate the approach makes sense. My only comment is about the limited selection of the baselines.
The authors in the related work reasonably exclude methods that do not preserve output consistency but mention approaches like SpecInfer (Miao et al., 2023b), Medusa (Cai et al., 2024), and its extension HYDRA (Ankner et al., 2024). While orthogonal to AdaDecode, it could be interesting to discuss further or report their results to contextualize the contribution better.

**Other Comments Or Suggestions:**

Regarding the passage on deviation from theoretical guarantees, the authors hypothesise a possible cause originating from FP16 precision used at inference time.
It would be interesting to see some experiments performing inference tweaking the precision.

The results that show that generalist heads are significantly slower might suggest a limited applicability of the approach in a "production" setting.
It would be interesting to expand more on this in the discussion and suggest potential strategies to mitigate this effect.

**Other Strengths And Weaknesses:**

Already highlighted in previous sections.

**Questions For Authors:**

No important further questions besides the comments.

**Relation To Broader Scientific Literature:**

The paper's findings are extremely relevant, given the strong need to improve latency in LLM without increasing the already high hardware requirements. The work is timely and builds on existing concepts addressing key limitations.

**Theoretical Claims:**

Checked the proof for Lemma A.1.
The proof is clear, but I invite the authors to expand on why P* high-rank implies E* is full rank, as this seems speculative. It would be interesting to show an empirical study on the models considered for the experiments in the same appendix.

---

> ### Author Rebuttal · Authors · 2025-04-01
>
> We sincerely thank the reviewer for the insightful comments and appreciation of our work.
>
> **[Q1]: While approaches like SpecInfer (Miao et al., 2023b), Medusa (Cai et al., 2024), and its extension HYDRA (Ankner et al., 2024) are orthogonal to AdaDecode, it could be interesting to discuss further to contextualize the contribution better.**
>
> **[A1]**: We agree that these tree-based speculative decoding methods are orthogonal to AdaDecode, but we believe it is possible and interesting to explore how our method could potentially be combined with tree-based decoding techniques. For instance, one potential approach could be generating multiple tokens at intermediate layers using our lightweight LM heads, rather than introducing additional full LM heads as in Medusa and always drafting tokens at the last layer. This strategy could allow for a more efficient draft token generation while maintaining the benefits of tree-based verification, and we believe this will be an interesting direction to explore in future work.
>
>
> **[Q2]: It would be interesting to show an empirical study on the models to show E\* is full rank.**
>
>
> **[A2]**: Thanks for the great suggestion. Please refer to Table 1 in this [PDF](https://anonymous.4open.science/r/AdaDecode-ICML2025-132E/rebuttal/To_Reviewer_2V7R_and_KXLJ.pdf).
>
>
> **[Q3]: The paper and appendix clearly describe experimental designs and analyses. I tried to run the code, but at the current stage, some needed artifacts are missing (e.g., adadecode/llama_3.1_8b_instruct)**
>
>
> **[A3]**: Please find the datasets and model checkpoints at this anonymous repo: https://huggingface.co/AnonyResearcher
>
>
> **[Q4]: Regarding the passage on deviation from theoretical guarantees, the authors hypothesise a possible cause originating from FP16 precision used at inference time. It would be interesting to see some experiments performing inference tweaking the precision.**
>
> **[A4]**: Following the reviewer’s suggestion, we re-run the experiments in Figure 4 with FP32. While FP32 offers slightly higher numerical consistency than FP16, it still does not reach 100% consistency.
> |Method|SpecDec|Self-SpecDec|LookAhead|SWIFT|AdaDecode|
> |-|-|-|-|-|-|
> |FP16|0.97|0.98|0.98|0.98|0.99|
> |FP32|0.98|0.98|0.99|0.98|0.99|
>
> This aligns with [the finding in the open-source community](https://github.com/huggingface/transformers/issues/30413) that the output of speculative decoding can differ slightly from standard decoding due to numerical precision inaccuracies and minor variations in token probabilities during computation (please refer to this [discussion](https://github.com/huggingface/transformers/issues/25420#issuecomment-1775317535) for a detailed study on the impact of different precisions). We would like to note that, during our experiments, we also noticed that different hardware specifications and library versions contribute to variances, demonstrating that half-precision is not the only factor affecting consistency.
>
> **[Q5]: The results that show that generalist heads are significantly slower might suggest a limited applicability of the approach in a "production" setting.**
>
> **[A5]**: While we acknowledge the advantage of using a generalist head that can support a mix of domain requests, we would like to highlight the growing trend of developing specialized models for domain-specific use cases.
> In many real-world production settings, instead of training a single monolithic model for all domains, specialized models offer greater efficiency and improved performance in their respective areas. For example, Cursor and GitHub Copilot are optimized for programming assistance, while models like Qwen-2.5-Math and  DeepSeek-Prover are designed for mathematical reasoning and theorem proving. In such applications, only a specific task type is considered, making a domain-specific LM head more preferable.
> That said, we believe our method can also produce good generalist LM heads. We hypothesize that the lower performance observed in the ablation using a generalist head is primarily due to the relatively small size of our mixed-domain dataset (<20K examples). With more extensive mixed-domain training data, we expect the performance of the generalist head to improve significantly. In this work, we focus on demonstrating that lightweight training (<2 GPU hours) is sufficient to produce high-quality intermediate-layer LM heads and achieve substantial speedups with our method. Exploring more comprehensive training for generalist heads will be interesting future work.
>
>
> We hope the reviewer finds our response helpful. We are also happy to incorporate additional suggestions you might have!  Thanks again for your time!

---

> > ### Comment · Reviewer_KXLJ · 2025-04-03
> >
> > I thank the authors for addressing all my comments and I reflected this in my updated score.

---

> > > ### Author Response · Authors · 2025-04-03
> > >
> > > We are glad that our response addressed the reviewer’s concerns. Thank you again for the support and constructive comments, and we will include them in our revision accordingly.

---

### Official Review · Reviewer_u7As · 2025-03-14

**Overall Recommendation:** 1

**Summary:**

The authors propose to use early exiting to accelerate autoregressive decoding in LLMs while leveraging adaptive layer parallelism for efficient hardware deployment. The early exiting framework uses a lightweight head at intermediate layers to enable high-confidence early token predictions. An additional verification step is also added to ensure early-predicted tokens match the results of standard autoregressive decoding. AdaDecode achieves upto 1.73X speed up in token generation on several token generation tasks such as summarization, codegen, and mathematical reasoning.

**Claims And Evidence:**

The authors provide multiple experiments to support the claims in the paper.

**Essential References Not Discussed:**

The paper seems to miss many of the existing related works in the literature with identical/significant overlap:

[1] Bae, Sangmin, et al. "Fast and robust early-exiting framework for autoregressive language models with synchronized parallel decoding." arXiv preprint arXiv:2310.05424 (2023).

[2] Elhoushi, Mostafa, et al. "LayerSkip: Enabling early exit inference and self-speculative decoding." arXiv preprint arXiv:2404.16710 (2024).

[3] Liu, Jiahao, et al. "Speculative decoding via early-exiting for faster llm inference with thompson sampling control mechanism." arXiv preprint arXiv:2406.03853 (2024).

[4] Cai, Tianle, et al. "Medusa: Simple llm inference acceleration framework with multiple decoding heads." arXiv preprint arXiv:2401.10774 (2024).

[5] Varshney, Neeraj, et al. "Accelerating llama inference by enabling intermediate layer decoding via instruction tuning with lite." arXiv preprint arXiv:2310.18581 (2023).

**Experimental Designs Or Analyses:**

Please refer to the weaknesses section. While the experimental results are valid, they lack comparison to the many prior Early-Exiting works.

**Methods And Evaluation Criteria:**

The benchmarks and speedup comparisons  between the proposed method and the baselines are meaningful.

**Other Comments Or Suggestions:**

None

**Other Strengths And Weaknesses:**

While the authors have done a good job in comparing their method to approaches other than early exiting, such as SpecDecode and Swift, they have not performed the obvious comparison to the numerous publications on early exiting in the literature.

- The method offers no novelty compared to prior work and it is unclear what the authors are contributing. Plugging in an intermediate head for early exiting inside a language model is not a new contribution.

- Combining verification / speculative decoding with early exiting is not a new idea and has been published in the references provided above.

- The related work section on early exiting just mentions a few works and refers the reader to a survey. The authors should clarify how their method is differentiated from prior work.

**Questions For Authors:**

None

**Relation To Broader Scientific Literature:**

The paper does not provide any new contributions compared to prior methods on early exiting for autoregressive decoding and fails to mention or compare to them. At its current state, the paper provides no novelty or new contribution to the field.

**Theoretical Claims:**

The authors provide a proof in Lemma A.1 in the appendix which appears valid. The assumption that $P^*$ is high rank makes sense. Additionally, as indicated, since $E^*$ is full rank, matrix $E^{(i)}$ can be expressed as a linear transformation of $E^*$.

---

> ### Author Rebuttal · Authors · 2025-04-01
>
> We thank the reviewer for the thoughtful comments.
>
> **[Q1]: The method offers no novelty compared to prior work and it is unclear what the authors are contributing.**
>
> **[A1]**: We would like to highlight that our work provides an efficient solution to tackle the key limitations of speculative decoding and early exiting through our technical innovations.
>
> Speculative decoding and early exiting have notable limitations: (1) speculative decoding typically incurs substantial training and memory costs due to the need for an additional draft model, and (2) early exiting is unable to leverage off-the-shelf intermediate layer representations without requiring further training or modifications, which can cause output deviations from standard autoregressive decoding.
>
> To address these challenges, we propose the following key innovations:
>
> - **Leveraging off-the-shelf intermediate layer representations**: For the first time, we demonstrate that, despite the challenges of using off-the-shelf intermediate layers for next-token prediction (Fig. 3a), learning a simple linear projection on these features yields surprisingly good next-token prediction (Fig. 3b). This finding has significant practical implications: The full fine-tuning of the entire model (e.g., LayerSkip [3]) or the introduction of a full additional layer and full LM head (e.g., EESD [4]) may not be necessary. Learning a simple linear projection based on frozen intermediate-layer features significantly reduces not only training complexity but also facilitates our design objective of guaranteeing output parity to vanilla decoding.
>
> - **Efficient lightweight LM heads without loss of expressiveness**: Through theoretical analysis in Lemma A (which is unknown in prior works), we prove that our lightweight intermediate LM heads (i.e., the transformation matrices) are lossless proxies for full LM heads, which reduce memory costs by 30x. This innovation allows us to serve multiple lightweight LM heads at various intermediate layers (otherwise introducing multiple full LM heads will cause expensive training and memory costs), enabling dynamic early exiting and delivering significant speedups (as shown in the “w/ fixed-layer early prediction” ablation of Table 2).
>
> To better position our contributions, we present a comparison table that highlights how our contributions differ from recent works. Please refer to Table 1 in this [PDF](https://anonymous.4open.science/r/AdaDecode-ICML2025-132E/rebuttal/To_Reviewer_u7As.pdf).
>
>
> **[Q2]: While the authors have done a good job in comparing their method to approaches other than early exiting, they have not performed the obvious comparison to the numerous publications on early exiting in the literature.**
>
> **[A2]**: To address the reviewer’s concern,  we would like to make the following clarifications.
>
> - **Why not compare with output-inconsistent methods (e.g,. FREE [1], LITE [2], LayerSkip [3])**: Our work is explicitly designed under the constraint of output consistency—producing the same outputs as standard autoregressive decoding. Relaxing this constraint leads to fundamentally different problem settings and design objectives, making direct comparisons with output-inconsistent methods inappropriate.
>
> - **Why not compare with tree-based decoding methods (e.g,. SpeccInfer [8], Medusa [5])**: Tree-based decoding methods are orthogonal to early exiting methods: The former generates multiple tokens simultaneously, accelerating **horizontally (across time steps)**, while the latter reduces per-token computation by parallelizing deep layers after early exit, accelerating **vertically (within each time step's forward pass)**. A direct comparison between these two orthogonal approaches would conflate their respective benefits. As precedent, SWIFT [7], LayerSkip [3], or Self-SpecDecode [6] also didn't compare with tree-based decoding methods for the same reason.
>
> If you find our response satisfactory, we would be grateful if you could consider raising your score. Thanks again for your time!
>
> **References**
>
> [1] Bae et al. Fast and robust early-exiting framework for autoregressive language models with synchronized parallel decoding. 2023
>
> [2] Varshney et al. Accelerating llama inference by enabling intermediate layer decoding via instruction tuning with lite. 2023
>
> [3] Elhoushi et al. LayerSkip: Enabling early exit inference and self-speculative decoding. 2024.
>
> [4] Liu et al. Speculative decoding via early-exiting for faster llm inference with thompson sampling control mechanism. 2024
>
> [5] Cai et al. Medusa: Simple llm inference acceleration framework with multiple decoding heads. 2024
>
> [6] Zhang et al. Draft & verify: Lossless large language model acceleration via self-speculative decoding. 2023
>
> [7] Xia et al. SWIFT: On-the-Fly Self-Speculative Decoding for LLM Inference Acceleration. 2024
>
> [8] Miao et al. Specinfer: Accelerating large language model serving with tree-based speculative inference and verification. 2024

---

> > ### Comment · Reviewer_u7As · 2025-04-09
> >
> > I thank the authors for their response, however, my concerns regarding lack of novelty and lack of comparisons to a plateura of efficient decoding methods remain.

---

### Official Review · Reviewer_2V7R · 2025-03-18

**Overall Recommendation:** 3

**Summary:**

The paper proposes to improve autoregressive decoding process in LLMs by decoding by using intermediate layer outputs when the confidence is high. A lightweight LM head is trained to decode the next token from such intermediate layer output enabling this method to be applied on pretrained models without requiring retraining from scratch. Remaining layer computations are executed in parallel with subsequent tokens as needed. The approach shows speedup over baselines in experiments.

## update after rebuttal

The authors have clarified my concerns around some of the technical details of the work and therefore I am increasing my score to 3. I still feel that the novelty is a bit limited and my concerns around the rank of the transformation matrix $E^*$ and the cost of domain specific heads have not been fully alleviated, but I would not be opposed to accepting this paper as it does add to the discussion in the literature on adaptive inference.

**Claims And Evidence:**

The authors claim that $E^{(i)}$, the lightweight LM head at layer $i\ \forall i$, used to check the exit condition can be represented as a linear transformation of the last layer LM head $E^*$. However, the proof presented in Appendix A requires $E^*$ to be full rank but does not conclusively demonstrate that that is the case. The authors only show that the rank of $E^*$ is lower bounded by the rank of a matrix $P$ that is likely to be high rank. However, this is neither a deterministic nor a probabilistic guarantee on $E^*$ being full rank.

**Essential References Not Discussed:**

N/A

**Experimental Designs Or Analyses:**

I checked the experiments in the main paper and do not have any issues with them.

**Methods And Evaluation Criteria:**

Yes

**Other Comments Or Suggestions:**

I would suggest saying that "$E^*$ is likely to be full rank", as opposed to "$E^*$ must be full rank" in Appendix A and modifying the corresponding claim in Section 2.1 accordingly.

**Other Strengths And Weaknesses:**

Strengths:

1. The observation that the existing LM head does not work well with intermediate layer outputs and that training an LM head for intermediate layers will work enables this process to work with pretrained models which is a very useful feature in the era of foundation models and a clear improvement over prior works that require training from scratch.

2. The efficiency modifications for minimizing the complexity of the new LM head and using adaptive layer parallelism to compute KV cache on the fly can help reduce inference cost.

Weaknesses:

1. The work appears to be a synthesis of ideas from the early exit and speculative decoding areas and so the overall novelty is a bit limited.

2. The proof of Lemma A.1 does not appear to be entirely correct (as mentioned above) and this weakens the argument that representing the intermediate LM heads as a linear projection of the final LM head will never lead to a loss of expressivity.

3. Certain technical details appear to be inconsistent (see questions below)

4. It is mentioned in Section 4.3 that domain specific heads for intermediate layers leads to significantly faster generation than heads trained on a mix of domain. However, this will significantly increase the memory/cost requirements for this approach in real LLM-based services where requests come from a mix of domains/users.

**Questions For Authors:**

1. How can layer 2 of t2 and t3 run in parallel (Fig 2) when the 2nd layer of t3 is depended on the output (KV cache) from the second layer of t2?

2. How will adding $t_i$ (token generated by the intermediate layer) to the parallel processing list of deeper layers in line 235 help in parallel processing? Don't we need to add the layer output instead to compute the outputs of subsequent layers?

3. Shouldn't there be a condition after line 238 for resuming processing from any rejected token $t$, as described after equation (2), after the final layer is reached at any position?

**Relation To Broader Scientific Literature:**

The paper builds upon prior work in speculative decoding and early exits. Unlike early exit which skips the computation of KV cache in post exit layers, this work performs those computations in parallel with subsequent tokens. Unlike speculative decoding which requires a separate drafter model for generating outputs this work uses the same model for both generation (intermediate layers) and verification (last layer).

**Theoretical Claims:**

I checked the proof of Lemma A.1 in Appendix A and believe that it isn't entirely correct as described above.

---

> ### Author Rebuttal · Authors · 2025-04-01
>
> We appreciate the reviewer’s insightful feedback.
>
> **[Q1]: I would suggest saying that "$E^\star$ is likely to be full rank", as opposed to "must be full rank" in Appendix A and modifying the corresponding claim in Section 2.1 accordingly.**
>
>
> **[A1]**: We provide an empirical validation confirming that $E^*$ is indeed full rank in this [PDF](https://anonymous.4open.science/r/AdaDecode-ICML2025-132E/rebuttal/To_Reviewer_2V7R_and_KXLJ.pdf).
>
> Moreover, as indicated by Lemma 1 of (Yang, 2018), the language modeling problem can be completely solved (i.e., achieve a 0 loss) if $\text{rank}(P) < d$. However, this is practically infeasible due to the inherent complexity of natural language, suggesting that $\text{rank}(P) > d$.
>
> We’ll clarify our theoretical claim in the revision for improved precision.
>
> **[Q2]: The work appears to be a synthesis of ideas from the early exit and speculative decoding areas and so the overall novelty is a bit limited.**
>
> **[A2]**: To better position our contributions, we provide a comparison table (Table 2) in the same PDF. We also clarify our novelty in Q1 to Reviewer u7As.
>
> **[Q3]: Domain-specific heads for intermediate layers will significantly increase the memory/cost requirements for this approach in real LLM-based services where requests come from a mix of domains/users.**
>
> **[A3]**: While we acknowledge the advantage of a generalist head, we would like to highlight the growing trend of developing specialized models for domain-specific use cases such as programming (e.g., Cursor/Github Copilot) and mathematical reasoning (e.g, Qwen-2.5-Math). In such applications, only a specific task type is considered, making a domain-specific LM head more preferable.
>
> That said, we believe our method can also produce good generalist LM heads. We hypothesize that the lower performance observed in the ablation using a generalist head is primarily due to the relatively small size of our mixed-domain dataset (<20K examples). With more extensive mixed-domain training data, we expect the performance of the generalist head to improve significantly. In this work, we focus on demonstrating that lightweight training (<2 GPU hours) is sufficient to produce high-quality intermediate-layer LM heads and achieve substantial speedups with our method. Exploring more comprehensive training for generalist heads will be interesting future work.
>
> **[Q4]: How can layer 2 of t2 and t3 run in parallel (Fig 2) when the 2nd layer of t3 is dependent on the output (KV cache) from the second layer of t2?**
>
> **[A4]**: While it is true that, in standard autoregressive decoding, the 2nd layer of token $t_3$ depends on the KV cache from the 2nd layer of token $t_2$, our method eliminates this dependency by generating the next token earlier. As illustrated in Fig 2, $t_3$ is produced using only the 1st layer output of $t_2$, enabling the 2nd-layer KV caches of $t_2$ and $t_3$ to be computed in parallel. This is analogous to prefilling, where knowing multiple tokens upfront allows for simultaneous KV cache calculation across those tokens. This parallel computation is the core mechanism behind our method's efficiency.
>
> **[Q5]: How will adding $t_i$ (token generated by the intermediate layer) to the parallel processing list of deeper layers in line 235 help in parallel processing? Don't we need to add the layer output instead to compute the outputs of subsequent layers?**
>
> **[A5]**: Line 235 updates $\mathcal{P}$, the list of tokens that need to be processed at each layer. For instance, if $t_1$ is an early prediction at layer 8, its KV cache at layers 9-32 remains empty. Then for the next token $t_2$, we have the following adaptive parallelism strategy to calculate their KV caches:
> - Phase 1: The first 8 layers of $t_2$ are processed individually
> - Phase 2: The remaining layers 9-32 of $t_2$ are processed in parallel with $t_1$
>
> Specifically, at the beginning of Phase 2, we concatenate the hidden representation of $t_1$ at layer 8 with that of $t_2$ at layer 8 to enable parallel processing. These intermediate layer outputs (hidden representations) are managed accordingly when updating $\mathcal{P}$. Since the KV cache computation automatically requires the layer outputs, such operations on layer output are omitted in the algorithm for simplicity.
>
> **[Q6]: Shouldn't there be a condition after line 238 for resuming processing from any rejected token $t$, as described after equation (2), after the final layer is reached at any position?**
>
> **[A6]**: Indeed, this "resume processing" operation is internally handled in Line 240 – if any rejection happens, Line 240 will remove disgarded tokens from $y$ and add the replacement token to $y$ so that the generation resumes from the correct token, as described in Section 2.2. We will update the annotations in Algorithm 1 to clarify this.
>
> If you find our response satisfactory, we would be grateful if you could consider raising your score. Thanks again for your time!

---

### Decision · Program_Chairs · 2025-05-01

**Decision:**

Accept (poster)

**Comment:**

The paper proposes an approach to speed up LLM inference by combining early-exiting with speculative decoding. The core idea is to allow intermediate layers to predict the next token, with these predictions subsequently being verified in parallel via the speculative decoding mechanism. This guarantees quality-neutrality, while allowing for speed ups owing to a reduction in the amount of sequential processing.

Reviewer opinion on the work was mixed. On the one hand, reviewers found the method to be sound and intuitive, with a convincing set of empirical results and ablations. On the other hand, some reviewers raised concerns around novelty compared to a large number of existing works also touching upon similar ideas. It was also requested that there be empirical comparisons to some of these works.

The author response argued that the the present work _is_ novel compared to existing works, which either forego quality-neutrality by not making use of speculative decoding, or which do not combine speculative decoding with depth-wise adaptivity.

Owing to the disagreement in scores, the AC also reviewed the paper, and some of the prior references suggested by reviewers. Our reading is as follows:
- there are several works which introduce some form of early-exiting, but without explicit quality-neutrality guarantees (e.g., [1, 2]). These are certainly topically relevant, but the present paper's focus on quality-neutrality is a reasonable distinction.
- there are several works which leverage speculative decoding, with innovations in how the drafting is performed (e.g., [3]). These are certainly topically relevant, but the present paper's focus on drafting based on an intermediate layer is a reasonable distinction.
- LayerSkip [4] and EESD [5] both consider a combination of early-exiting and speculative decoding. Both papers _are_ cited in the present work; however, we believe a far **more detailed discussion** of their similarities and differences is needed.
- from our reading of [4], we believe that Section 4.3 discusses a highly similar idea to the present paper, wherein intermediate-layer drafts are verified via the speculative decoding process. However, a distinction is in the mechanism used for _training_ the underlying model: [4] proposes a layer dropout strategy, which allows for re-use of the final unembedding matrix by any intermediate layer. This requires fine-tuning of the _entire_ network, which can result in performance differences to the original network. By contrast, the present work attaches additional trainable parameters at each layer, while the backbone is _frozen_. This allows for quality preservation of the original network performance.
- from our reading of [5], we believe that again the core idea illustrated in Figure 2 is highly similar to the present paper. However, three distinctions are the use of a _fixed_ layer to perform drafting, rather than a dynamic layer selection in the present paper; the use of Thompson sampling to allow for a dynamic number of draft tokens, rather than a fixed selection in the present paper; and the use of a separate Transformer model on top of the early layer embeddings, as opposed to a lightweight linear layer. The last point in particular means that the present proposal involves far fewer parameters than [5].
- from the above, we believe the present paper _can be reasonably argued to offer novelty_ over [4, 5]; and that these novelties can have practical implications (specifically, freezing of the backbone to guarantee quality neutrality, and use of lightweight linear layers). This said, per above, it is really incumbent on the authors to state such arguments clearly; we believe these are not sufficiently spelled out in the original submission, though the response did touch upon these.
- finally, with the above understanding, one question is whether or not an empirical comparison is necessary. For [4], we believe that a comparison to the layer dropping training strategy is of considerable interest; in the settings that the present paper considers, does this actually lead to a significant quality drop compared to the original network? For [5], we believe that a comparison of the linear head versus a Transformer head would be of considerable interest, and could drive home the point of parameter efficiency of the present method (assuming one ends up with similar acceptance rates). We agree that the Thompson sampling strategy is somewhat orthogonal, and so a comparison may not be strictly necessary.

Overall, on balance we believe the paper does have some distinctiveness to the existing literature, albeit it may still be argued to synthesize a few different ideas (which is not necessarily a bad thing). The paper must however make the discussion of these contributions far more detailed. Further, an empirical comparison to the key strategies one of [4, 5] would really help drive home the advantages of the proposed approach.

[1] Bae, Sangmin, et al. "Fast and robust early-exiting framework for autoregressive language models with synchronized parallel decoding." arXiv preprint arXiv:2310.05424 (2023).

[2] Varshney, Neeraj, et al. "Accelerating llama inference by enabling intermediate layer decoding via instruction tuning with lite." arXiv preprint arXiv:2310.18581 (2023).

[3] Cai, Tianle, et al. "Medusa: Simple LLM inference acceleration framework with multiple decoding heads." arXiv preprint arXiv:2401.10774 (2024).

[4] Elhoushi, Mostafa, et al. "LayerSkip: Enabling early exit inference and self-speculative decoding." arXiv preprint arXiv:2404.16710 (2024).

[5] Liu, Jiahao, et al. "Speculative decoding via early-exiting for faster llm inference with thompson sampling control mechanism." arXiv preprint arXiv:2406.03853 (2024).